# Sperm sequencing reveals extensive positive selection in the male germline

Matthew D. C. Neville[1], Andrew R. J. Lawson[1], Rashesh Sanghvi[1], Federico Abascal[1], My H. Pham[1], Alex Cagan[1], Pantelis A. Nicola[1], Tetyana Bayzetinova[1], Adrian Baez-Ortega[1], Kirsty Roberts[1], Stefanie V. Lensing[1,2], Sara Widaa[2], Raul E. Alcantara[1,3], María Paz García[4], Sam Wadge[4], Michael R. Stratton[1], Peter J. Campbell[1], Kerrin Small[4], Iñigo Martincorena[1], Matthew E. Hurles[1] & Raheleh Rahbari[1✉]

Mutations that occur in the cell lineages of sperm or eggs can be transmitted to offspring. In humans, positive selection of driver mutations during spermatogenesis can increase the birth prevalence of certain developmental disorders[1–3]. Until recently, characterizing the extent of this selection in sperm has been limited by the error rates of sequencing technologies. Here we used the duplex sequencing method NanoSeq[4] to sequence 81 bulk sperm samples from individuals aged 24–75 years. Our findings revealed a linear accumulation of 1.67 (95% confidence interval of 1.41–1.92) mutations per year per haploid genome driven by two mutational signatures associated with human ageing. Deep targeted and exome NanoSeq[5] of sperm samples identified more than 35,000 germline coding mutations. We detected 40 genes (31 newly identified) under significant positive selection in the male germline that have activating or loss-of-function mechanisms and are involved in diverse cellular pathways. Most of the positively selected genes are associated with developmental or cancer predisposition disorders in children, whereas four of the genes exhibited increased frequencies of protein-truncating variants in healthy populations. We show that positive selection during spermatogenesis drives a 2–3-fold increased risk of known disease-causing mutations, which results in 3–5% of sperm from middle-aged to older individuals with a pathogenic mutation across the exome. These findings shed light on germline selection dynamics and highlight a broader increased disease risk for children born to fathers of advanced age than previously appreciated.

All human cells accumulate mutations throughout life. In proliferating tissues, acquired driver mutations that confer a selective advantage can promote the expansion of individual clones in competing stem and progenitor cell populations. Although patterns of selection and clonal expansion have been extensively studied in cancers, recent research has also highlighted their occurrence in normal tissues during ageing[6–14].

Spermatogonial stem cells of the testis occupy a distinct niche relative to other studied normal tissues. Among replicating cells, these stem cells have the lowest mutation rate, which is around 5–20-fold lower than other studied somatic cell types[12]. They are also the only adult proliferating cells with the potential to transmit mutations to offspring, balancing self-renewal and spermatogenesis to produce 150–275 million sperm per day[15,16]. Targeted sequencing studies have revealed that driver mutations are acquired in spermatogonial stem cells and that these cell populations expand along seminiferous tubules, which results in increased fractions of mutant clones that are detectable in sperm[1,17–20]. Notably, all the male germline drivers identified so far are activating missense hotspot mutations, which contrasts with the broader range of activating and inactivating drivers observed in cancers and somatic tissues. These germline driver mutations can have substantial implications for offspring, as

such mutations are found in a set of 13 genes all known to cause severe developmental disorders[3]. This effect leads to a significant increase, up to 1,000-fold, in the sporadic birth prevalence of these disorders, with a strong correlation between the age of the father and prevalence[2].

Technical limitations, related to the polyclonality and low mutation rate of testis and sperm, have prevented extensive characterization of this selection beyond a limited set of genes. Recent advances in error-corrected duplex DNA sequencing approaches use information from both DNA strands to detect mutations at single-molecule resolution[21–24]. These new methods have proven successful for the accurate estimation of mutation burden in sperm, with an error rate of $<5 \times 10^{-9}$ per base pair[4]. Here we combine the duplex approaches of whole-genome NanoSeq[4] with deep whole-exome and targeted NanoSeq[5] to characterize positive selection in the male germline and to quantify its consequences for the accumulation of disease mutations in sperm.

## Cohort and sequencing coverage

We performed whole-genome NanoSeq[4] of bulk semen samples ($n = 81$, 1–2 time points per donor, 57 donors with an age range of 24–75 years)

[1]Cancer, Ageing and Somatic Mutation, Wellcome Sanger Institute, Hinxton, UK. [2]Sequencing Operations, Wellcome Sanger Institute, Hinxton, UK. [3]Quotient Therapeutics, Saffron Walden, UK. [4]Department of Twin Research and Genetic Epidemiology, Kings College London, London, UK. ✉e-mail: rr11@sanger.ac.uk

and matched blood (n = 119, 1–3 time points, 63 donors with an age range of 22–83 years) from men in the TwinsUK cohort[25] (including 8 monozygotic and 3 dizygotic twin pairs; Methods and Supplementary Table 1). The analysed semen samples had sperm counts >1 million per ml, as samples below this threshold showed evidence of somatic cell contamination (Extended Data Fig. 1 and Supplementary Note 1). The mean number of unique DNA molecules per site where a mutation was callable (duplex coverage (dx)) was 3.7 dx in sperm and 4.3 dx in blood (Extended Data Fig. 2a). For sperm (a haploid cell), 1 dx is equivalent to one cell, whereas for blood (a diploid cell), 2 dx is equivalent to one cell.

## Mutational burden and signatures

We identified single nucleotide variants (SNVs) and small insertion–deletion mutations (indels) in whole-genome NanoSeq data from sperm and blood. Inherited germline variants were excluded using matched blood (Methods). From the 6,653 SNVs detected in sperm (Supplementary Table 2), we estimated an accumulation of 1.67 substitutions per year per haploid genome (95% confidence interval (CI) = 1.41–1.92, linear mixed-effect regression). This result is comparable with estimates from paternal de novo mutations (DNMs) in family pedigrees[26] of 1.44 substitutions per year (95% CI = 1.00–1.87) and seminiferous tubules of the testes[12] of 1.40 substitutions per year (95% CI = 1.02–1.76; Fig. 1a). Indels accumulated in sperm at a rate of 0.10 per year per haploid genome (95% CI = 0.06–0.15), again similar to the rate observed in DNMs[26] of 0.08 haploid indels per year (95% CI = −0.02 to 0.17) and testes[12] of 0.08 haploid indels per year (95% CI = 0.02–0.13; Fig. 1b and Extended Data Fig. 3a).

From 92,035 SNVs and 4,641 indels detected in whole blood (Supplementary Table 3), we estimated an accumulation of 19.9 substitutions per year per diploid genome (95% CI = 17.3–22.5; Fig. 1c) and 0.9 indels per year (95% CI = 0.7–1.1; Fig. 1d and Extended Data Fig. 3b). Both estimates are within the range of mutation rates observed for specific cell types in blood[13] and are consistent with measuring a weighted average of these cell types in whole blood (Extended Data Fig. 4a,b). Individuals had a mean of 7.6-fold more substitutions per base pair per year (range of 4.2–11.5; Fig. 1e) and 6.3-fold more indels per base pair per year (range of 2.2–18.7; Fig. 1e) in blood than in sperm. Accounting for twin status or multiple time points from the same individuals had a significant predictive effect for mutation burden in blood but not in sperm (Supplementary Note 2).

The SNV mutational signatures in sperm were inferred to be SBS1 (mean 16%) and SBS5 (mean 84%), which were the expected clock-like ageing signatures[27] (Fig. 1f,g). In blood, SBS1 (mean 15%) and SBS5 (mean 75%) were also the main mutational signatures, with an additional contribution of SBS19 (mean 10%), which has been linked to persistent DNA lesions in haematopoietic stem cells[27] (Fig. 1f,g). We observed that all signatures were correlated with age (Extended Data Fig. 4c,d). SBS1 and SBS5 accumulated in individuals at a mean of 8.9-fold (range of 2.3–39.1) and 6.8-fold (range of 3.7–10.9) higher rate in blood than in sperm, respectively (Extended Data Fig. 4e). This finding indicates that SBS19 does not explain a substantial fraction of the mutation burden gap between the two tissues.

## Selective pressure dynamics in sperm

We next investigated positive selection in protein-coding regions in sperm. We used a capture-based modification to NanoSeq[5] to deeply sequence coding regions from the same set of semen samples. Specifically, we sequenced 38 samples using whole-exome NanoSeq to a mean depth of 551 dx per sample (20,923 cumulative dx). We also sequenced 81 samples using targeted NanoSeq to a mean depth of 985 dx per sample (79,811 cumulative dx) with a target panel that consisted of 263 canonical cancer driver genes, 107 of which are also associated with developmental disorders (Methods, Extended Data

Fig. 2a and Supplementary Table 4). We detected 56,503 (58% in coding regions) SNV and indel mutations from the exome panel and 5,059 (58% in coding regions) from the targeted cancer panel (Methods and Supplementary Table 5). The mutation burdens for exome and targeted samples were consistent with whole-genome NanoSeq after correcting for the relative trinucleotide composition of sequencing coverage (Extended Data Fig. 2b).

The majority of variants (99.5%) were detected only in a single duplex molecule of a sample. Similarly, in the 23 samples with 2 time points (mean of a 12.1-year gap), 99.3% of the 5,143 variant calls from the first time point were not called in the second time point. These results are consistent with sperm being a highly polyclonal collection of cells derived from a large population of spermatogonial stem cell progenitors in the testis.

The exome-wide strength of positive selection in sperm was quantified by estimating the rate of nonsynonymous (N) relative to neutral synonymous (S) mutations (dN/dS ratio, where dN/dS = 1.0 indicates neutrality). We used the dNdScv algorithm[28] with modifications to account for duplex sequencing coverage per base, CpG methylation levels in testis samples and pentanucleotide context. These modifications refined exome-wide dN/dS ratios by resolving specific mutation rate biases but had minor effects on gene-level dN/dS ratios (Extended Data Fig. 5 and Supplementary Note 3).

Using this model, the dN/dS ratio in the exome-sequenced samples were estimated to be 1.07 (95% CI = 1.04–1.10). This ratio implies that 6.5% (95% CI = 3.8%–9.1%) of the observed nonsynonymous substitutions in sperm conferred a clonal advantage during spermatogenesis in this cohort. Splitting the cohort into thirds by age, the exome-wide dN/dS ratio was 1.01 (95% CI = 0.93–1.09) in 26–42 year olds, 1.03 (95% CI = 0.97–1.10) in 43–58 year olds and 1.09 (95% CI = 1.06–1.13) in 59–74 year olds (Fig. 2f). A linear regression of dN/dS ratio against age group showed a positive trend, although significance was not reached (P = 0.18). This result suggests that the dN/dS ratio may increase over the male lifespan. If so, then the cohort-wide dN/dS ratios presented here partially reflect the age distribution of samples (age range of 26–74 years, mean of 53 years).

We next compared the dN/dS ratios across gene sets related to spermatogenesis expression[29] (Fig. 2g). Genes with the highest dN/dS ratios were those that were highly expressed during spermatogenesis (1.25, 95% CI = 1.13–1.38) and most specific to differentiated spermatogonial stem cells (1.11, 95% CI = 1.05–1.17). By contrast, the genes that were not expressed in spermatogenesis (0.98, 95% CI = 0.88–1.11) and the genes most specific to elongating spermatids (1.01, 95% CI = 0.94–1.08) showed dN/dS ratios close to neutrality. These results are consistent with the understanding that excess nonsynonymous mutations observed in sperm confer a competitive advantage earlier in their cell lineage, specifically in the spermatogonial stem cells of the testis[18].

## Driver gene discovery

We then combined exome and targeted panel datasets to investigate which genes were driving the signal of positive selection (Methods). We applied dN/dS tests for excess nonsynonymous mutations at gene-wide and SNV hotspot levels, which together identified 40 genes under significant positive selection. Of these, 35 genes reached exome-wide significance at the gene level (false discovery rate q < 0.1; Supplementary Table 6) and/or contained 1 of 17 exome-wide significant hotspots (q < 0.1; Extended Data Table 1). The genes *PTPN11*, *MIB1*, *RIT1*, *FGFR3*, *EP300* and *FGFR2* were significant in both the gene and hotspot tests. The genes *KDM5B*, *NF1*, *SMAD6*, *CUL3*, *RASA2*, *PRRC2A*, *PTEN*, *ROBO1*, *DDX3X*, *CSNK2B*, *KRAS*, *PPM1D*, *ARID1A*, *BRAF*, *HRAS*, *KMT2E*, *SCAF4*, *BMPR2*, *TCF12*, *CCAR2*, *DHX9*, *NSD1*, *LZTR1*, *ARHGAP35*, *CBL*, *SSX1* and *RBM12* were significant in only the gene test. The genes *SMAD4* and *FAM222B* were significant in only the hotspot test (Fig. 2a,b).

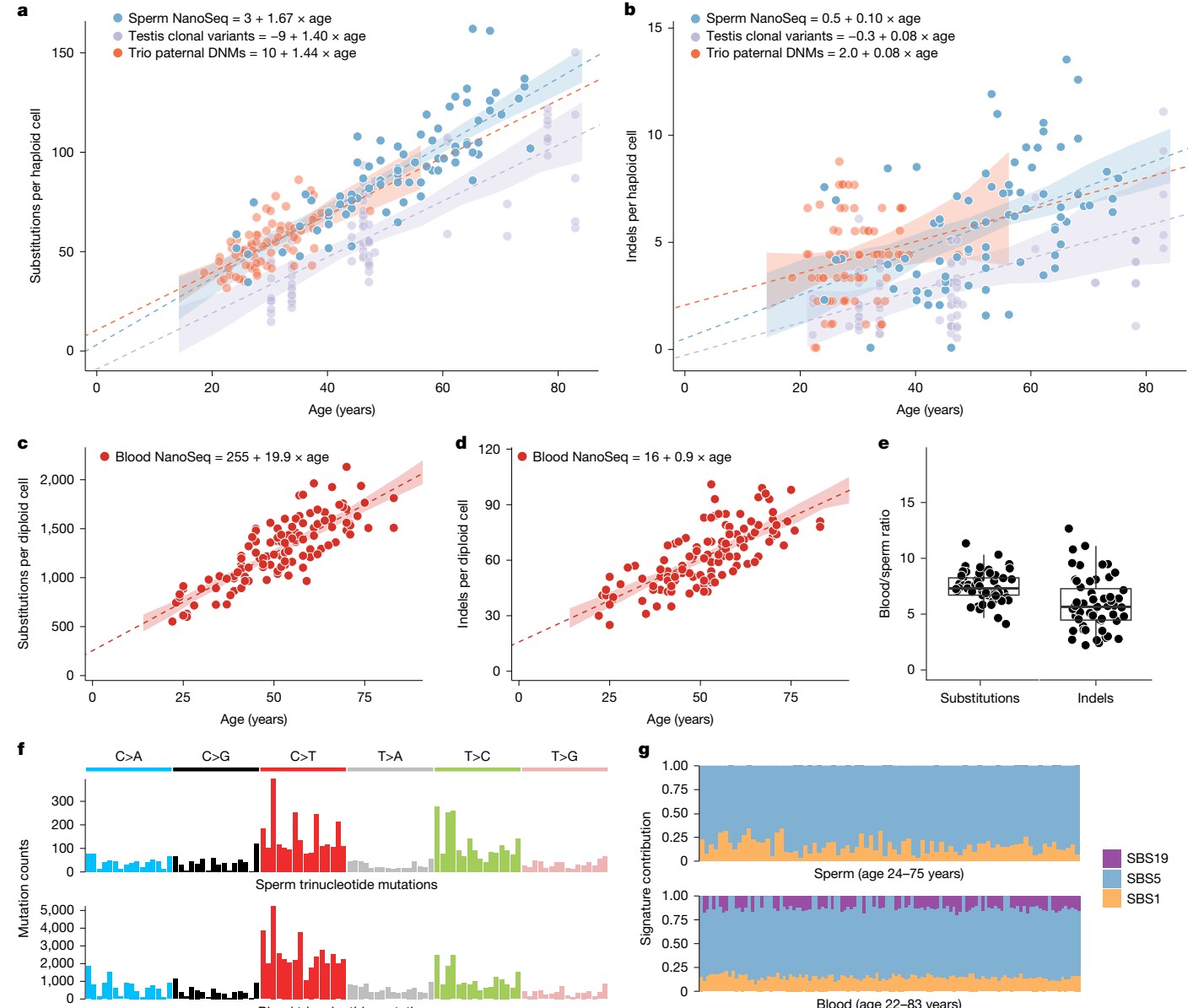

**Fig. 1 | Mutational burden and signature analysis in sperm and matched blood. a,b**, Substitutions (**a**) and indels (**b**) per haploid cell from whole-genome NanoSeq of sperm, trio paternal DNMs[26] called with standard sequencing and clonal variants from seminiferous tubules of testis[12] called with standard sequencing. Dots indicate single donors, except for testis with 1–15 samples per donor. **c,d**, Substitutions (**c**) and indels (**d**) per diploid cell for different ages from blood NanoSeq samples. **e**, Ratio of blood to sperm substitutions and indels per diploid cell per year. Each dot corresponds to an individual with both a blood and sperm sample. For individuals who had multiple time points, the mean value of all time points in that tissue was used. Box plots show the median as the centre line, the 25th and 75th percentiles as box limits and whiskers extending to the largest and smallest values within 1.5× the interquartile range from the limits from $n = 57$ biologically independent samples. **f**, Trinucleotide mutation counts in all sperm and blood samples. **g**, Contributions of the signatures SBS1, SBS5 and SBS19 in sperm and blood samples ordered by age. For **a**–**d**, models are linear mixed regressions, with the central line showing the model fit and the shaded bands indicating 95% CIs calculated using parametric bootstrapping.

We excluded *SEMG1* despite it reaching gene-level significance, as its extreme expression in seminal vesicles[30], lack of expression in spermatogenesis and indel-specific enrichment indicate indel hypermutation[31,32] from minor seminal vesicle DNA contamination rather than germline selection.

Subsequently, we carried out restricted hypothesis dN/dS tests at the per-gene and per-site level. The gene-level test examined only the set of 263 canonical cancer driver genes on our target panel. The site-level test used a set of 1,963 sites consisting of known cancer hotspots and recurrent DNM sites from the Deciphering Developmental Disorders (DDD) cohort[33]. This analysis identified five additional genes—*KDM5C*, *KMT2D*, *AR*, *CTNNB1* and *RAF1*—and seven hotspots not already significant at

the exome-wide level ($q < 0.1$; Fig. 2a,b, Extended Data Table 1 and Supplementary Table 7).

The genes previously implicated in germline positive selection all operate through activating missense mutations, with 12 linked to the RAS–MAPK signalling pathway[3] and one (*SMAD4*) linked to TGFβ–BMP signalling[34]. Our findings replicated *SMAD4* and 8 out of the 12 RAS–MAPK pathway genes as under significant positive selection in this dataset. The four genes that did not reach significance (*MAP2K1*, *MAP2K2*, *SOS1* and *RET*) each had between twofold and fourfold enrichment of missense mutations, which corresponded to nominally significant missense enrichment in all four genes ($P < 0.1$). Given the direct evidence for these genes driving clonal

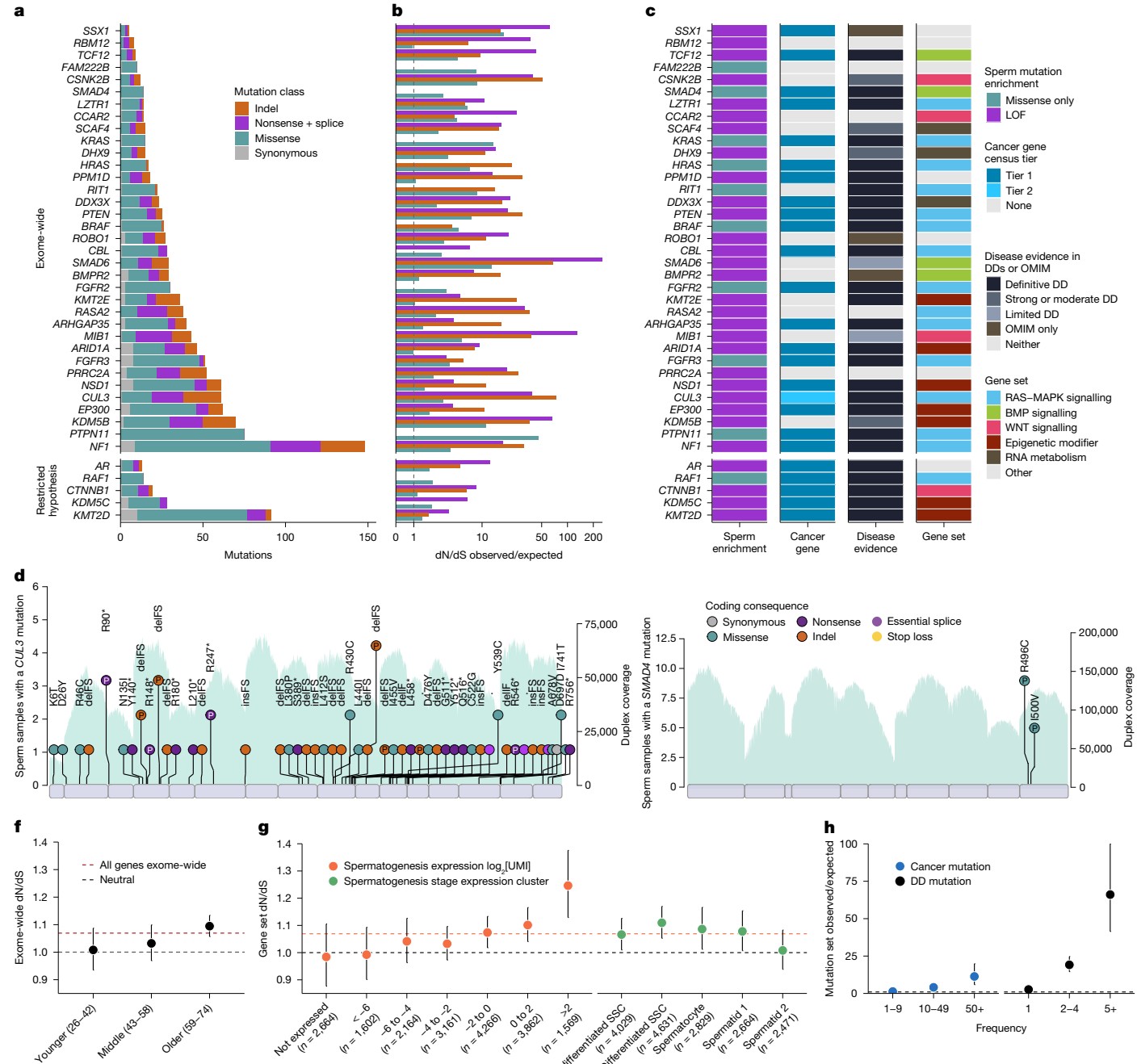

**Fig. 2 | Germline positive selection. a–c,** Genes with significant dN/dS ratios from exome-wide and restricted hypothesis tests. **a,** Mutation count split by mutation class. **b,** Enrichment over expectation of mutation classes. **c,** Mutation type driving dN/dS enrichment, COSMIC cancer gene tier, developmental disorder (DD) gene status in DDG2P and potential germline selection gene set. **d,e,** Observed sperm mutations across the cohort for *CUL3* and *SMAD4*. The height of each lollipop represents the number of biologically independent samples with a mutation at that position, and the colour indicates mutation type. Mutations are labelled with their amino acid consequence or, for insertions (ins) and deletions (del), as in-frame (IF) or frameshift (FS). A 'P' indicates pathogenic/ likely pathogenic classification in ClinVar. Exons are shown as purple rectangles and the blue background shows the total duplex coverage across the cohort. Lines below the gene indicate somatic mutations in cancer from COSMIC. **f,g,** dN/dS ratios for sperm SNVs across sets of individuals or genes, where the dotted black line indicates neutrality and the dotted orange line represents the cohort average across all genes. **f,** Exome-wide dN/dS ratios for younger, middle and older age groups of $n = 11$, $n = 9$ and $n = 18$ biologically independent sperm samples, respectively. **g,** Expression levels as $\log_2$ of unique molecular identifier (UMI) counts and cell-type clusters from single-cell sequencing of germ cells[29]. Germ-cell types include undifferentiated and differentiated spermatogonial stem cells (SSCs), spermatocytes, round spermatids (spermatid 1) and elongating spermatids (spermatid 2). Data are from mutations detected in $n = 38$ biologically independent sperm samples. **h,** Observed/ expected mutation rates in sperm for variant recurrence bins (data from COSMIC and DDD databases) using mutations from $n = 38$ biologically independent sperm samples. Data in **f–h** show ratio point estimates, error bars indicate 95% CIs.

selection in testis and nominal enrichment from sperm sequencing, we anticipate that each will reach exome-wide significance with deeper sequencing.

We estimate that together, the 44 genes linked to germline selection here or in previous studies contain 357 (95% CI = 319–387) excess nonsynonymous variants in exome sequenced sperm samples.

This result would account for 23% (95% CI = 14%–43%) of the total estimated driver variants across the exome. The wide CIs and the sensitivity of this estimate to the mutation model used (Supplementary Note 3) suggest that small uncertainties in mutation rates, when propagated across the exome, make it difficult to precisely estimate the fraction of drivers explained. Nevertheless, the findings indicate that additional driver genes remain to be discovered.

The 31 newly identified genes demonstrate that germline positive selection is not restricted to activating mutations or to the RAS–MAPK pathway. For instance, 30 out of the 31 genes are enriched for loss-of-function (LOF) mutations such as nonsense, splice and indel variants, which are indicative of protein-inactivating mechanisms of selection (Fig. 2c,d). Splitting the germline-selection genes and known cancer genes on the basis of ten canonical cancer pathways[35], we found that the top gene groups enriched in dN/dS are RAS–MAPK, WNT and TGFβ–BMP signalling (Extended Data Fig. 6a). Indeed, many of these selection genes are linked to the RAS–MAPK pathway (such as *NF1*, *CUL3* and *LZTR1*), WNT signalling (*CSNK2B*, *MIB1* and *CCAR2*) and TGFβ–BMP signalling (*SMAD6*, *TCF12* and *BMPR2*) (Fig. 2c). We also identified several genes that encode epigenetic modifiers (*KDM5B*, *KDM5C*, *ARID1A*, *KMT2D*, *KMT2E*, *EP300* and *NSD1*) and genes encoding RNA metabolism proteins (*DHX9*, *DDX3X* and *SCAF4*). These findings highlight a new diversity of genes, mutational mechanisms and pathways that drive germline selection.

It has been observed that cancers and germline developmental disorders share causal pathways and genes[36,37]. Notably, the 13 genes previously linked to germline positive selection are all known cancer and known developmental disorders genes[3,34]. This pattern holds, but to a lesser extent, in the new germline selection genes identified here: 16 out of 31 genes are tier 1 or 2 cancer census genes[38] and 27 out of 31 are linked to monogenic disorders in the Development Disorder Genotype–Phenotype (DDG2P) or Online Mendelian Inheritance in Man (OMIM) databases[39] (Fig. 2c and Supplementary Table 8).

The overlap between germline positive selection and cancer or developmental disorders is also apparent at the variant level. Somatic mutations that are most frequently observed (>50 times) in the Catalogue Of Somatic Mutations In Cancer (COSMIC) database are enriched 11-fold (95% CI = 6–20) among our sperm mutation dataset after adjusting for the expected mutation rate (Methods). Similarly, germline mutations that are most frequently observed (>5 times) in a large cohort of children with developmental disorders are enriched 66-fold (95% CI = 41–100) in our sperm mutation dataset (Fig. 2h). Moreover, the mutation types enriched in sperm for a given gene largely matched those enriched in cancer and linked to developmental disorders (Extended Data Figs. 6a and 7). These results show a clear overlap between genes, hotspots and mutation mechanisms that drive germline positive selection, cancer and developmental disorders. A notable exception to this pattern is *SMAD4*, which has two distinct missense hotspots in sperm that are developmental disorder hotspots causal for Myhre syndrome[40] but are not often seen in cancers, a result that replicates recent findings[34] (Fig. 2e).

## Enrichment of disease-causing mutations in sperm

Given the association of many positively selected genes to disease, it is of interest to assess to what degree positive selection may increase the fraction of sperm with potential disease-causing mutations, and thus the birth prevalence of the associated disease. To estimate the fraction of sperm with specific classes of variants, we aggregated the variant allele frequencies (VAFs) of different mutation types and compared this to expected values from our dNdScv mutation model (Methods).

The fraction of sperm carrying noncoding or synonymous mutations increased linearly with age, as predicted by the neutral model (Extended Data Fig. 8). By contrast, missense, truncating and coding indel variants deviated above expected in older individuals, an observation consistent with the dN/dS results and indicative of age-related positive selection.

To assess potential disease burden, we compiled a conservative list of probable monoallelic disease-causing mutations, including pathogenic or likely pathogenic variants from ClinVar[41] and highly damaging variants (combined annotation dependent depletion (CADD)[42] value of >30) in high-confidence monoallelic developmental disorder genes from the DDG2P cohort[39]. Across all ages, the observed frequency of disease mutations exceeded expectation from the germline mutation model. The expected fraction ranged from 0.73% in 30 year olds to 1.6% in 70 year olds. When fitting a quasibinomial regression, the observed values ranged from 2% (95% CI = 1.6%–2.5%) in 30 year olds to 4.5% (95% CI = 3.9%–5.2%) in 70 year olds, with a significant relationship between the observed fraction and age ($P = 1.75 \times 10^{-5}$; Fig. 3c). These differences represent similar enrichments of 2.8-fold (95% CI = 2.2 to 3.5) and 2.9-fold (95% CI = 2.5 to 3.3) in 30 year olds and 70 year olds, respectively.

Notably, the disease cell-fraction estimates were made up of many low-frequency variants rather than being driven by individual high VAF mutations. The estimates in exome samples are made up of a mean of 18.3 distinct variants (range of 4–62) per individual. Furthermore, 692 out of 696 (99.4%) of all those variants were only observed in a single sperm, similar to the average of all variants (99.5%).

We next investigated to what degree the observed enrichment of disease mutations can be attributed to driver mutations in positively selected genes. Fitting a quasibinomial regression, we observed a strong positive correlation between age and driver rate ($P = 7.95 \times 10^{-6}$) with an estimated 0.5% (95% CI = 0.3%–0.8%) of sperm from individuals at age 30 years and 2.6% (95% CI = 2.0%–3.3%) of sperm from individuals aged 70 years with a known driver mutation (Fig. 3b). However, only about two-thirds (65.6%) of those driver mutations met our criteria of likely disease-causing.

Mutations in sperm that are likely disease-causing and those classified as known driver mutations represented overlapping, yet distinct, annotations (Fig. 3a). Across the cohort, an estimated 3.3% of sperm carried a likely disease-causing mutation. Of this, approximately one-third (1.2%) was expected by the neutral mutation model, another third (1.1%) was explained by known driver mutations and the remaining third (1.0%) was unexplained by either source. These findings indicate that the increase in likely disease-causing mutations is largely driven by germline positive selection on known genes, but also indicate that additional driver genes with disease associations remain to be identified.

After examination of the driver mutations that did not meet our criteria for likely disease-causing, we estimated that they affected 0.6% of sperm across the cohort. The inheritance consequences of these mutations are unclear. For instance, *SMAD6* variants, which are linked to variably penetrant congenital phenotypes[43], may be disease-causing in some cases but not others. Other possibilities include mutations that are unannotated disease-causing, impair fertilization, are embryonic lethal or act through biallelic disruption.

A large fraction of sperm with disease and/or driver mutations could be attributed to a small number of genes. Of the 374 genes with at least one such variant, 33 with ≥5 independent mutations, most under significant positive selection (26 out of 33), accounted for 42.8% of the disease or selection burden (Fig. 3d). Notably, six of those genes, all under significant positive selection (*KDM5B*, *MIB1*, *SMAD6*, *PRRC2A*, *NF1* and *PTPN11*) alone explained over 20% of the disease or selection fraction. This result suggests that although all individual driver mutations were observed at low allele frequencies, they were disproportionately concentrated in specific driver genes.

We next examined whether risk factors beyond age contribute to the accumulation of disease or driver mutations in sperm. Known germline mutagens include chemotherapy, inherited DNA repair defects[33,44] and weaker influences from genetic ancestry and smoking[45]. Although our cohort did not include individuals with chemotherapy exposure or DNA repair defects, phenotype data on body mass index (BMI), smoking and

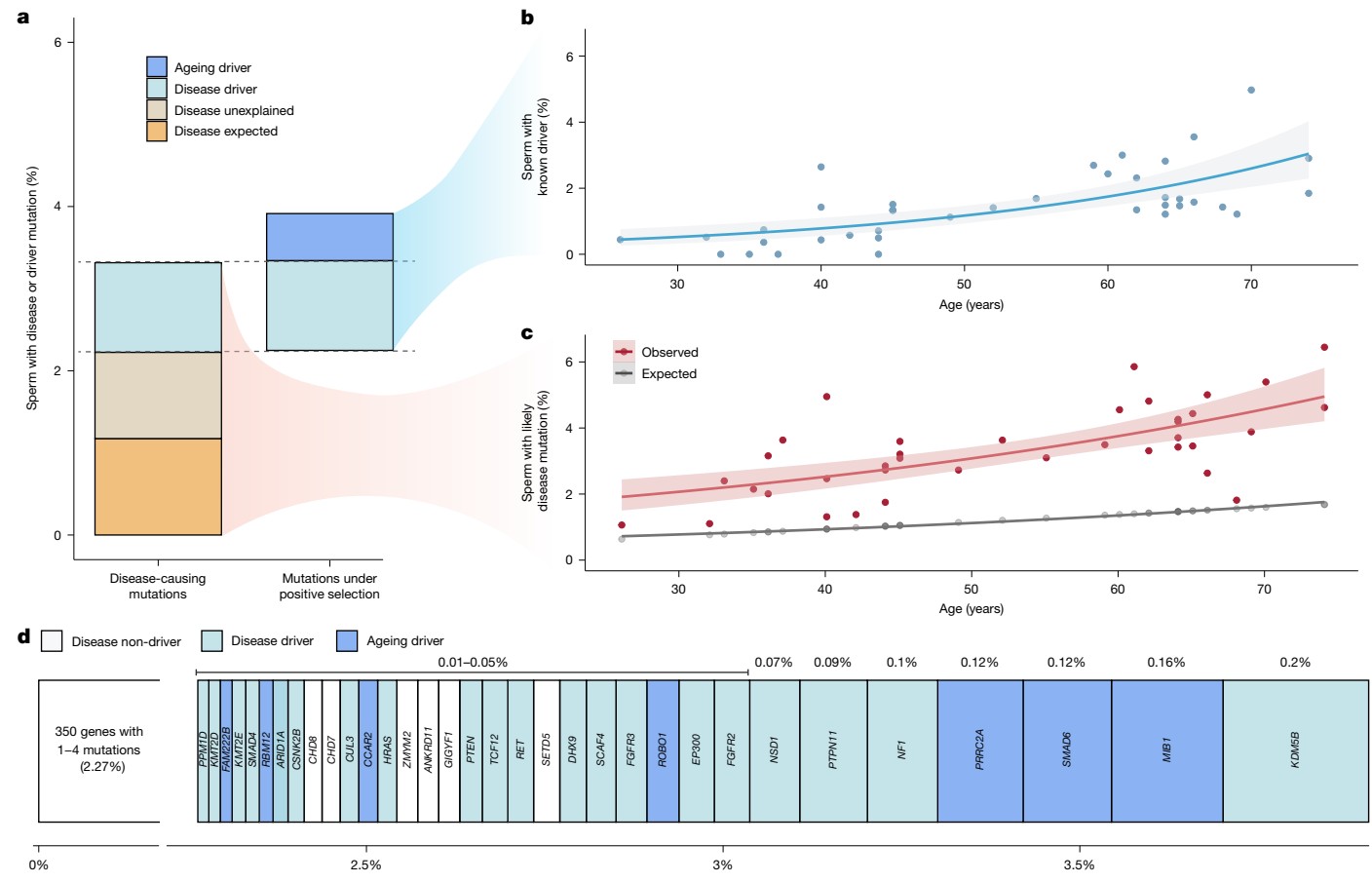

**Fig. 3 | Pathogenic burden. a**, Estimated mean percentage of sperm in the cohort with a likely monoallelic disease mutation (left) or a driver mutation in a germline-selection gene (right). Disease mutations are divided into the fraction that was expected from the mutation model, the portion explained by driver variants and the portion unexplained. Driver mutations are split by those contributing to the disease mutations and the remainder, 'ageing drivers'. **b**, Estimated percentage of sperm per individual with a driver mutation by age. **c**, Observed and expected percentages of sperm with a likely disease mutation by age. For **b** and **c**, the central line represents the model-predicted mean from quasibinomial regression, and shaded bands indicate 95% CIs. **d**, Cohort means from **a** split by gene and ordered by estimated mutation percentage. Per-gene contributions are shown above each gene; the summed contributions of all genes are shown below. Genes with four or fewer variants are grouped on the left with a condensed *x* axis for clarity.

alcohol consumption, all known to affect mutation burden in some somatic tissues[46], were available. Using multivariate generalized linear models (Methods and Extended Data Fig. 9), we tested associations between these factors and measures of mutation burden, signatures and driver cell fractions, correcting for multiple testing. Across sperm datasets (targeted, exome and whole genome) no significant effects were found for BMI, smoking pack-years or alcohol drink-years. Notably, in blood samples, smoking and alcohol consumption showed significant effects on SNV and SBS5 mutational signature burdens. These findings indicate that the male germline may be largely protected from these exposures, although further investigation in larger and more diverse cohorts is needed.

## Selection across germline variant sources

Mutations in sperm account for about 80% of DNMs and are therefore also the primary origin of population-level variants. Comparisons of these germline mutation sources offer insight into how positive selection in spermatogonia manifests over time.

Control DNMs from unaffected offspring[47] showed a dN/dS ratio near neutrality (0.98, 95% CI = 0.90–1.08; Fig. 4a), whereas DNMs from children with developmental disorders showed a marked nonsynonymous enrichment (1.36, 95% CI = 1.33–1.39; Fig. 4a), a result consistent with previous reports[33]. However, ascertainment biases in these

cohorts probably distort the observed ratios, and larger, unbiased birth cohorts will be required to resolve the baseline of DNMs entering the population.

To assess how selection shapes germline variants in the population, we analysed dN/dS ratios across allele frequencies in gnomAD[48] (Fig. 4a). This analysis revealed a decay in dN/dS ratios at higher allele frequencies, a result consistent with purifying selection across generations. Together, these results suggest that positive selection is the greater force that acts on germline mutations during spermatogenesis, whereas negative selection predominates over generations. This finding mirrors the well-established contrast between selection on cancer and population-level mutations[28].

Most genes in gnomAD have positive *z* scores, which indicates LOF depletion due to negative selection. Of the 31 selection genes with LOF enrichment in sperm (range of 3-fold to 50-fold), 27 showed LOF depletion in gnomAD, a result consistent with positive selection during spermatogenesis but purging in the population by negative selection. Notably, four genes, *SMAD6*, *MIB1*, *LZTR1* and *SSX1*, had more LOFs than expected in gnomAD (Fig. 4b). The latter three are among the strongest LOF-enriched outliers in gnomAD (v.2) and are flagged by gnomAD for unexplained LOF enrichment. Our results suggest that their increased LOF frequency in gnomAD reflects increased input from germline positive selection, with insufficient negative selection to remove them from the population.

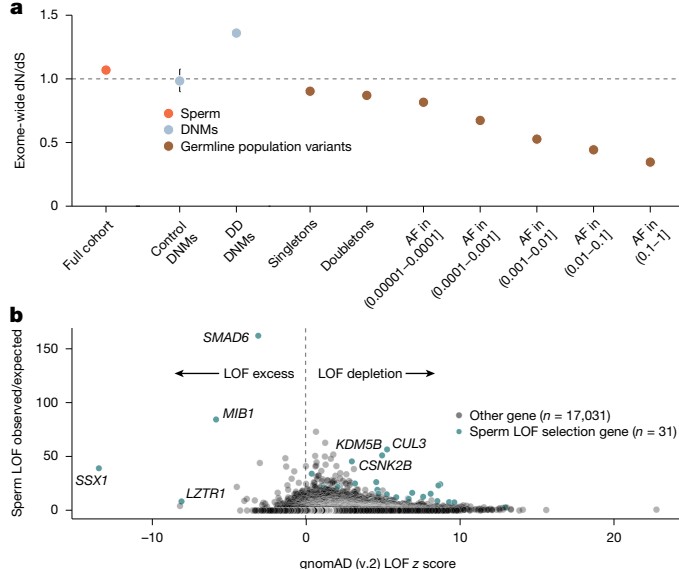

**Fig. 4 | Comparison with population variation. a**, Exome-wide dN/dS ratio point estimates across different variant sets, including *n* = 38 biologically independent sperm samples, DNMs from *n* = 1,886 healthy trios[47] and *n* = 22,742 trios from the DDD cohort[33], and population variants from *n* = 125,748 individuals in gnomAD[48], split by allele frequency (AF). Error bars indicate 95% CIs. **b**, Observed/expected enrichment of missense and LOF (essential splice, nonsense or indel) variants in positively selected genes in sperm (*x* axis) from dN/dS models versus gnomAD (v.2) LOF *z* scores. Positive *z* scores indicate LOF depletion, whereas negative scores indicate excess over expected.

## Discussion

We sequenced sperm and blood from healthy men spanning a wide age range to quantify mutation rates and positive selection in the male germline. Mutation rates and signatures in sperm matched those from family trio and testis studies[12,26,49–52]. Despite sharing SBS1 and SBS5 signatures, mutations accumulated around eightfold more slowly in sperm than blood, a result that supports our previous observations of the protected nature of the germline relative to the soma[12].

By analysing more than 35,000 coding mutations from sperm exome-wide, we corroborated key findings from previous studies of spermatogonial selection[1,17–20] and significantly advanced our understanding of its scope and impact. Our results replicated 9 out of 13 known germline-selection genes and confirmed that the impact of selection increases with age, leading to an increased fraction of sperm with pathogenic variants. We also identified 31 new positively selected genes, thereby broadening the range of pathways and mechanisms implicated in this process. The discovery of diverse genes and the identification of LOF as a common selection mechanism highlight the power of NanoSeq to enable exome-wide searches and to detect mutations in single cells, overcoming the limitations of previous methods reliant on high-frequency gain-of-function missense mutations. These findings reveal that germline selection operates in the broader framework of cellular selection, driven by many of the same genes and mechanisms that shape clonal dynamics in somatic tissues. However, unlike somatic selection, germline selection affects offspring phenotypes and influences evolutionary trajectories.

Perhaps unsurprisingly then, driver mutation burden in sperm is low compared with many somatic proliferative tissues. In middle-aged to older men, we showed that only 1–3% of sperm carry a known driver mutation. Low levels have also been reported in colon (about 1%) and liver (1–5%)[8,9]. In blood, many individuals fell within this range, although some showed large expansions of single clones[14]. Much higher burdens are seen in epithelial tissues such as endometrium, oesophagus and skin, where 30–50% of cells often have driver mutations[6,7,10].

These differences are probably shaped in part by both tissue architecture and mutation rate. In the testis, the low mutation rate combined with the tubular structure may together be crucial to constrain the generation and spread of driver clones. This low driver rate also aligns with the rarity and biology of tumours that originate from spermatogonia, which are typically driven by aneuploidy rather than sequential driver mutations[53].

Our findings have significant implications for disease and evolutionary studies that rely on germline mutation models, as they do not currently account for positive selection. We showed that this selection can distort estimates of selective constraint based on population-level data. Germline selection will also affect the discovery of disease-causing genes from DNM enrichment tests. For example, excess LOF mutations in *MIB1* found in developmental disorder trios[33] probably reflects germline selection rather than disease association, as they are more common in population cohorts than expected and do not show phenotype correlation[54]. In principle, case–control tests for DNM enrichment would avoid this bias, but require large cohorts of age-matched controls for sufficient statistical power. Until such resources become available, DNM enrichment studies should consider evidence for germline selection influencing individual genes presented here (Supplementary Table 5).

Unlike *MIB1*, most genes under positive selection during spermatogenesis are linked to severe monogenic disorders. We demonstrated that positive selection in spermatogenesis led to a substantial twofold to threefold enrichment in sperm with likely disease-causing mutations across the age range studied. As a result, we estimate that 3–5% of sperm from men aged over 50 years carry a likely disease-causing mutation, a value that exceeds previous estimates based on germline mutation models[55].

Although these risks are substantial, typical paternal age at conception is lower than in our cohort, which suggests that the impact on birth outcomes may be more modest. Moreover, the relationship between sperm mutations and birth prevalence remains uncertain. Many pathogenic variants in sperm may not result in live births owing to impaired fertilization, embryonic lethality or pregnancy loss (Supplementary Note 4). Nevertheless, growing awareness of these risks may prompt interest in reproductive planning, genetic counselling or clinical interventions. Translating these findings into specific clinical recommendations, however, presents challenges. Unlike inherited variants, which are present in every cell, or aneuploidies, for which clinical focus is limited to a few recurrent events, pathogenic mutations in sperm are both highly diverse and individually rare, which makes them difficult to target with standard screening approaches. Nonetheless, targeted risk assessment may be valuable in specific contexts, such as individuals with an increased risk of sperm hypermutation due to impaired DNA repair or chemotherapy exposure[52,56].

Our results focused on SNVs and indels, without capturing other variant classes. Although long-read sequencing studies have begun to explore these in sperm[57] and trios[58], further work is needed to capture the full range of human germline variation and its impact on offspring.

This study provides important insights into the historically underexplored reproductive risks of ageing in men. Unlike the well-established effects of maternal ageing on oocyte quality and aneuploidy[59], we are only beginning to understand the full scope of male-specific risk, including the clonal expansion of driver mutations. As trends towards delayed reproduction continue[60], it is essential to recognize that the age of both parents contributes to risk of disorders in offspring. Future research will refine our understanding of selective pressures and disease risk associated with germline mutation, clarifying their implications for human reproduction and health.

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

# Methods

## Ethics

This study was carried out under TwinsUK BioBank ethics, approved by the North West–Liverpool Central Research Ethics Committee (REC reference 19/NW/0187), IRAS ID 258513 and earlier approvals granted to TwinsUK by the St Thomas' Hospital Research Ethics Committee, later the London–Westminster Research Ethics Committee (REC reference EC04/015).

## Sample collection

Semen samples were collected or obtained from archival samples with informed consent from 75 research participants in the TwinsUK cohort[25]. Archival whole-blood samples were also obtained from 67 of those men from the TwinsUK BioBank. A total of 104 semen samples spanned an age range of 24–75 years and included 29 men with 2 time points separated by a mean of 12.1 years (range of 12–13 years) and the remaining 46 men with a single time point. A total of 133 blood samples were collected at an age range of 22–83 years from men with 1–4 time points. The mean interval between blood time points was 8.1 years (range of 1–13 years). In the cohort, there were a total of nine monozygotic twins and three dizygotic twin pairs. Counts of samples, time points and twin pairs that were successfully sequenced and passed analysis quality control thresholds are summarized in Supplementary Table 1.

## Metadata

Self-reported age, height, weight, ethnicity, twin zygosity, smoking and alcohol consumption were obtained from questionnaires provided by TwinsUK and periodically taken. All individuals who provided ethnicity information indicated white. BMI was calculated as $weight/height^2$. A smoking pack-years was defined as 365 packs of cigarettes and a total pack-year was calculated using the highest estimate across all questionnaires from cigarettes per day or week and total years smoked. Alcohol drink-years was calculated using average weekly alcohol consumption extrapolated to the duration of adult life before sampling (age – 18).

## DNA extraction

DNA was extracted from sperm samples using a Qiagen QIAamp DNA Blood Mini kit. Isolation of genomic DNA from sperm protocol 1 (QA03 Jul-10) was followed, but with the exceptions of substituting DTT in place of β-mercaptoethanol for buffer 2 and substituting buffer EB in place of buffer AE for the elution of DNA.

DNA was extracted from whole blood using a Gentra Puregene Blood kit using the protocol for 10 ml of compromised whole blood from the Gentra Puregene Handbook (v.06/2011).

## Targeted gene panels

Three separate Twist Bioscience gene panels were used for targeted NanoSeq sequencing in this study: (1) a custom pilot panel of 210 genes; (2) a similar but extended custom panel of 263 genes (Supplementary Table 4); (3) and a default exome-wide panel of 18,800 genes. The two custom gene panels were highly similar, with the extended panel being almost exclusively regions added to the pilot panel. From the 84 samples that underwent targeted sequencing, 13 were sequenced using a pilot panel of 210 canonical cancer or somatic driver genes, and all 84 were sequenced using the extended panel of 263 genes. Sequencing coverage and mutations were merged from samples sequenced on both targeted panels and they were treated as 'targeted' samples in the Article. The custom panels were designed by gathering sets of published lists of genes implicated as drivers in cancers[61–64] and somatic tissues[6,65] as previously described[5].

## Sequencing and library preparation

Restriction-enzyme whole-genome NanoSeq libraries were prepared as previously described[4] and subjected to whole-genome sequencing (WGS) at target 20–30× coverage on NovaSeq (Illumina) platforms to generate 150-bp paired-end reads with 9–10 samples per lane. Standard WGS of blood (31.7× median coverage) was used to generate matched-normal libraries for both restriction-enzyme NanoSeq blood and sperm samples.

Targeted and exome NanoSeq libraries were prepared by sonication and one to two rounds of pull down of target sequences. They were then PCR amplified and sequenced with NovaSeq (Illumina) platforms to generate 150-bp paired-end reads with 7–8 samples per lane for the targeted panel and 2 lanes per sample for the exome panel. These steps are described in detail in 'Sonication NanoSeq, Library amplification and sequencing, and Hybridization Capture' of supplementary note 1 of ref. 5.

## Base calling and filtering

All samples were processed using a Nextflow implementation of the NanoSeq calling pipeline (https://github.com/cancerit/NanoSeq). BWA-MEM[66] was used to align all sequences to the human reference genome (NCBI build37). Whole-genome NanoSeq samples were called with their matched WGS normal and default parameters except for var_b (minimum matched normal reads per strand) of 5 as needed for WGS normals, cov_Q (minimum mapQ to include a duplex read) of 15 and var_n (maximum number of mismatches) of 2.

For targeted and exome samples, we leveraged the high sequencing depth and high polyclonality to exclude variants with VAF > 10% instead of using a matched normal. Default parameters of the calling pipeline except for cov_Q of 30, var_n of 2, var_z (minimum normal coverage) of 25, var_a (minimum AS-XS) of 10, var_v (maximum normal VAF of 0.1) and indel_v (maximum normal VAF) of 0.1. After variant calling, we further filtered variants to those below 1% VAF. The few variants observed between 1% and 10% duplex VAF as variants were highly enriched for mapping artefacts, particularly for indels. No excluded variant from the additional 1% VAF threshold was found to be either a likely driver or a ClinVar pathogenic variant.

A set of common germline variants from dbsnp[67] and a custom set of known artefactual call sites in NanoSeq datasets were masked for coverage and variant calls as previously described[4].

## Assessing DNA contamination

The single-molecule accuracy of the duplex sequencing method NanoSeq allows sequencing of polyclonal cell types such as sperm, but also renders mutation calls sensitive to nontarget cell-type contamination and to contamination of foreign DNA. Nontarget cell-type contamination was evaluated using manual cell counting of semen samples, which resulted in the exclusion of six samples with a sperm count of <1 million sperm per ml. Sperm counting methods and analysis are detailed in Supplementary Note 1.

Foreign DNA contamination in whole-genome NanoSeq samples was assessed using verifyBamID[68], which checks whether reads in a BAM file match previous genotypes for a specific sample, with higher values indicating more contamination. Three blood whole-genome NanoSeq samples were excluded on the basis of a verifyBamID alpha value above the suggested[4] cut-off value of 0.005. In sperm, we found that several samples had outlier mutation burdens with verifyBamID values just below the 0.005 cut-off. This result is logical, as sperm has a much lower mutation rate compared with somatic tissues, for which the recommended cut-off was designed. Consequently, sperm samples are more sensitive to low levels of contamination. To account for this, we adjusted the verifyBamID alpha threshold for sperm to a more stringent level of 0.002, which resulted in the exclusion of three samples on the basis of this criterion.

When assessing foreign DNA contamination in targeted and exome samples, we found that 9 targeted and 3 exome samples had verifyBamID values above >0.002, slight outlier mutation burdens and a high ratio of SNP masked variants to passed variants (4-fold to 16-fold

## Article

more masked variants). After further investigation, we found that all samples that exceeded verifyBamID thresholds were processed in the same sequencing batch and that this contamination could be explained by inherited germline variants of other samples in that same batch. This result suggested that a small amount of cross-contamination may have occurred during sample preparation or sequencing steps. To remove contaminant germline mutations, we performed an in silico decontamination step as previously described[4]. This step involved calling germline variants from all targeted and exome samples using bcftools mpileup[69] at sites where there were >10 reads and a mutation call with VAF > 0.3. All such sites were subsequently masked across all samples for both mutation calls and coverage, which essentially extended the default common SNP mask to also include rare inherited variants across the cohort. This approach resulted in all samples previously identified as contaminated with mutation burdens consistent with their age and all with a ratio of masked to passed variants of <0.1, and were therefore retained for analysis.

### Corrected mutation burdens

Given that mutation rates are strongly influenced by trinucleotide composition, it is important to consider differences in sequence composition when comparing mutation rates in datasets that target different regions of the genome[70]. The coding region target panels and the restriction enzyme used in whole-genome NanoSeq for instance, each will systematically bias sequencing coverage to specific genomic regions that may not reflect the full genome. To correct for these effects, in each sperm NanoSeq dataset, we generated a corrected mutation burden relative to the full genome trinucleotide frequencies as previously described[4].

### Comparison of burden estimates

To compare whole-genome NanoSeq mutation burdens to mutation burdens from standard WGS, we multiplied the corrected mutation burden estimates described in the previous section by the genome size per cell type. We assumed 2,861,326,455 mappable base pairs in a haploid cell for germline datasets and the diploid equivalent for blood.

External datasets for comparison to NanoSeq results were processed to achieve comparable burden estimates. For testis WGS samples[71], we implemented a previously described method[4] that restricts analysis to regions with high coverage (20+ reads) that overlap with NanoSeq covered regions and corrected for differences in trinucleotide background frequencies relative to the full genome. For trio paternally phased DNMs from standard sequencing, as a callable genome size per sample following thorough filtering was available, we generated the mutation per cell estimate by multiplying the paternally phased DNM count by the ratio of the total genome size to the callable genome size of the sample as follows: DNMs paternal × (total genome/callable genome). For comparison with cell types in blood, we directly compared results to published mutation burden regressions[13].

### Mutation burden regressions

To investigate the relationship between age and mutation burdens, we performed linear mixed-effects regression analyses. For each tissue and mutation type for which a regression was performed, the model was constructed using the lmer function from the lme4 package[72] in R. Each model included age at sampling as a fixed effect and a random slope for each individual to account for multiple time point samples, with restricted maximum likelihood (REML) set to false, specified as follows:

$$\mathrm{lmer}(\mathrm{burden} \sim \mathrm{age} + (\mathrm{age} - 1|\mathrm{individual}), \ \mathrm{REML} = \mathrm{FALSE})$$

The 95% CIs for regression lines were calculated through bootstrapping by simulating prediction intervals. For each model, we generated 1,000 bootstrap samples. Predictions and their associated standard errors were calculated for a sequence of ages from 14 to 84 years. The 95% CIs were then derived by determining the range within which 95% of the bootstrap sample predictions fell.

### Mutational signature analysis

We extracted DNM signatures using hierarchical dirichlet process (HDP; https://github.com/nicolaroberts/hdp), which is based on the Bayesian hierarchical dirichlet process. HDP was run with double hierarchy: (1) individual identifier (ID) and (2) tissue types (either blood or sperm), without the COSMIC reference signatures[73] (v.3.3) as priors, on the mutation matrices. The number of mutations were normalized for the trinucleotide context abundance specific for each sample relative to the full genome. Both clustering hyperparameters, beta and alpha, were set to one. The Gibbs samples ran for 30,000 burn-in iterations (parameter 'burnin'), with a spacing of 200 iterations (parameter 'space'), from which 100 iterations were collected (parameter 'n'). After each Gibbs iteration, three iterations of concentration parameters were conducted (parameter 'cpiter'). Two components were extracted as DNM signatures, which were further reconstructed and decomposed into known COSMIC (v.3.3) SBS signatures using SigProfilerAssignment (https://github.com/AlexandrovLab/SigProfilerAssignment).

### Quantifying selection with dN/dS

To examine genes under positive selection and to quantify global selection, we used the dNdScv algorithm[28]. This algorithm was extended using the base-pair-level duplex coverage, the methylation level and the pentanucleotide context to capture more complex context-dependent mutational biases and to achieve higher accuracy for our selection analysis. Detailed methods for input mutations, model selection and evaluation, site dN/dS tests, driver mutation estimation and gene set enrichment are described in Supplementary Note 3.

### Gene disease and mechanism annotation

Positively selected genes were annotated with monoallelic disease consequences using the 29 February 2024 release of the DDG2P database[39] and the 21 June 2024 release of the OMIM database[74]. OMIM annotations related to somatic disease, complex disease or tentative disease associations were excluded.

Genes were also annotated for their potential mutation mechanism observed in sperm and cancer or developmental disorders when available. In sperm, genes were labelled as LOF if they had nominal enrichment of nonsense + splice variants and/or indel variants (ptrunc_cv <0.1|pind_cv <0.1) and 2+ LOF mutations. There were two exceptions to this, whereby genes met these thresholds but were labelled as activating owing to having a restricted repertoire of LOF mutations that are known to be oncogenic in cancers: *CBL* (LOFs in and downstream of the RING zinc finger domain)[75] and *PPM1D* (LOFs in final two exons)[76]. All other genes had missense enrichment only and were labelled as activating. The mechanism in cancer was defined by using the 'role in cancer' field of the COSMIC cancer gene census (v.99)[38], where 'oncogene' was labelled as activating and 'tumour suppressor gene' as LOF. Annotations of a fusion mechanism were not displayed, except for genes that had neither an oncogene nor a tumour suppressor annotation, which were labelled as 'fusion only'. The developmental disorder mechanism was defined by using the variation consequence field of DDG2P, for which 'restricted repertoire of mutations;activating' was labelled as activating and 'loss_of_function_variant' was labelled as LOF.

### Gene mutation plots

The lollipop gene mutation plots were created using a coordinate system, whereby position 1 was defined as the first coding base of the GRCh37-GencodeV18+Appris[77] transcript of the gene. The data sources included protein domains from UniProt[78], somatic mutations from the exome and genome-wide screens of the COSMIC (v.99)[38], ClinVar (release 2024.07.01)[41] pathogenic annotation, per base pair cohort-wide

(targeted + exome) NanoSeq coverage, sperm mutation count (number of independent individuals with a mutation) and mutation consequence and amino acid change annotated by the dNdScv algorithm[28]. These data were plotted using code modified from the lolliplot function in the trackViewer R package[79].

### Variant annotation

Variants were annotated using Ensembl's variant effect predictor (VEP)[80] with added custom annotations of mutation context, ClinVar (release 2024.07.01)[41], CADD (v.GRCh37-v1.6)[42] and the average methylation level in testis. Methylation data were obtained from whole-genome shotgun bisulfite sequencing methylation data from male testis samples from a 37-year-old (ENCFF638QVP) and 54-year-old (ENCFF715DMX) from the ENCODE project[71]. The average methylation level was calculated by selecting CpG sites with coverage of three or more and averaging the per cent of sites methylated between the two samples.

Variants were annotated as likely monoallelic disease-causing mutations if they met at least one of two criteria: (1) reported in ClinVar as pathogenic, likely pathogenic or if they were reported as 'conflicting classifications of pathogenicity', where the conflict was between reports of pathogenic/likely pathogenic and 'uncertain significance' with no reports of benign or likely benign and not specified as a recessive condition; or (2) were a highly damaging variant in high-confidence monoallelic developmental disorder genes from DDG2P[39]. Genes met the following criteria: (1) allelic requirement being monoallelic_autosomal, monoallelic_X_hem, monoallelic_X_het or mitochondrial; (2) confidence in strong, definitive or moderate; and (3) a mutation consequence of 'absent gene product'. Highly damaging was defined as being a 'high' impact variant in the VEP annotation (frameshift splice_acceptor, splice_donor, start_lost, stop_gained or stop_lost) or a missense variant with CADD[42] score >30 (top 0.1% damaging).

Variants were defined as a likely driver if they met the 'highly damaging' criteria defined above in a significant germline-selection gene with LOF mutation enrichment or if they were in 1 of the 24 significant mutation hotspots. This resulted in 320 variants being labelled as likely drivers in exome samples.

### Cell fraction mutation estimates

To calculate the mean count of synonymous, missense or LOF or pathogenic mutations per sperm cell, we summed the duplex VAFs of all variants in that class. For example, if an individual had 3 synonymous mutations each observed once with a duplex coverage of 100 at each of those sites, each of those variants would have a duplex VAF of 1/100 = 0.01. The sum of VAFs in this example would then be 0.03 and this would then be reported as the estimate for the mean count of synonymous variants per sperm cell for that individual. At low fractions such as 0.03, the mean count per cell is approximately equivalent to the percentage of sperm with this mutation class (3%); therefore, the driver and disease mutations are reported as percentage estimates. At higher fractions (for example, mean count > 1), the fractions are not equivalent to percentage, as many cells will have multiple variants of that class; therefore, the estimates are reported as mean count.

Expected mean counts for SNVs were generated by annotating each possible substitution at each covered site with an expected number of mutations per sample as given by expCountSNV = context_mut_rate × duplex_coverage × age_correction. The context_mut_rate was given by the 208 basePairCov + cpgMeth trinucleotide mutation model estimates for that trinucleotide + methylation mutation context (Supplementary Note 3 and Supplementary Table 9). Duplex coverage is the exact duplex coverage at that site for that sample. The age_correction parameter was given to normalize the mutation model estimates (derived from all exome samples) to the mutation rate of that sample based on age. Specifically, we fit a linear model to the mutation burden versus age of the exome samples and used this to generate a predicted mutation rate for each sample based on age. The per sample correction parameter was calculated as the age predicted mutation burden divided by the mean mutation rate of all exome samples ($3.89 \times 10^{-8}$). The resulting corrections spanned from 0.50 (youngest sample) to 1.42 (oldest samples). The expected indel mutation rate was calculated in the same way, except with a single mutation rate parameter (indels/bp) expCountIndel = indel_mut_rate × duplex_coverage × age_correction. The expected mean count was then calculated for each category (for example, synonymous or likely disease) by summing the expected values for each SNV and/or indel base pair matching the relevant annotation. As background for possible ClinVar pathogenic variants, we only considered indels of size 21 bp or less, the largest detected indel in the dataset. Regressions were fit with either linear models or generalized linear models (glm in R) with family = quasibinomial.

### Regression analysis

We tested for associations between mutation outcome variables from sperm genome, sperm exome, sperm targeted and blood genome NanoSeq data and the phenotype predictor variables of BMI, smoking pack-years and alcohol drink-years. These tests were performed using a Gaussian family generalized linear regression in R. For each mutation outcome variable the test took the following form: glm(mutationOutcome ~ age_at_sampling + BMI + pack_years + drinkYears, family = "gaussian").

The mutation outcome variables examined were SNV and indel burden from all four sequencing datasets, SBS1 and SBS5 count from sperm genomes, SBS1, SBS5 and SBS19 from blood genomes, and likely disease cell fraction and likely driver cell fraction from sperm targeted and sperm exomes. The significance of each predictor was assessed from the summary output coefficients of the model, and P values were adjusted for 68 total tests using the false discovery rate method.

### Reporting summary

Further information on research design is available in the Nature Portfolio Reporting Summary linked to this article.

### Data availability

Sequencing data are available through managed access at the European Genome-Phenome Archive (EGA) under accession numbers EGAD00001015591 (targeted and exome NanoSeq), EGAD00001015590 (whole-genome NanoSeq) and EGAD00001015592 (WGS matched normals). All non-TwinsUK files necessary to reproduce the results are available from GitHub (https://github.com/mattnev17/spermPositiveSelectionManuscript). Individual-level data from the TwinsUK cohort cannot be publicly shared or deposited owing to restrictions in the original participant consent. Access to these data is governed by the TwinsUK Resource Executive Committee (TREC), which reviews requests on a monthly basis. For information on how to apply for access to TwinsUK genotype and phenotype data, please see the website: https://twinsuk.ac.uk/researchers/access-data-and-samples/request-access/. This study also made use of the following publicly available resources: DDG2P (https://www.deciphergenomics.org/ddd/ddgenes); OMIM (https://omim.org); COSMIC Cancer Gene Census (https://cancer.sanger.ac.uk/census); gnomAD (v.2.1.1; https://gnomad.broadinstitute.org); ClinVar (https://www.ncbi.nlm.nih.gov/clinvar/); and bisulfite methylation data from the ENCODE Project (https://www.encodeproject.org).

### Code availability

The NanoSeq variant calling pipeline, available from GitHub (https://github.com/cancerit/NanoSeq; v.3.3), was used to process duplex sequencing data. R (v.4.3.1) was used for statistical analyses and visualization. Code modified from trackViewer (R/Bioconductor package v.1.38.0) was used to generate gene mutation lollipop plots.

lme4 (v.1.1-33) was used for linear mixed-effects models. ggplot2 (v.3.4.4) was used for plotting. dNdScv (https://github.com/im3sanger/dndscv; version as of commit on 29 September 2023) was used for selection analysis. HDP (https://github.com/nicolaroberts/hdp) was used for DNM mutational signature extraction. SigProfilerAssignment (v.0.0.27; https://github.com/AlexandrovLab/SigProfilerAssignment) was used for COSMIC signature fitting. All custom analysis code and scripts are available from GitHub (https://github.com/mattnev17/spermPositiveSelectionManuscript).

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

**Acknowledgements** We thank the TwinsUK research volunteers for participating in the study; L. O'Neill, C. Latimer and all of the CASM support team for their assistance; and C. Seymour, E. Ferraro and C. White for training and guidance on sperm counting. This research was funded by Wellcome Trust Grant 108413/A/15/D and 220540/Z/20/A. R.R. is funded by Cancer Research UK (C66259/A27114) and the Medical Research Council (MR/W025353/1). TwinsUK is funded by the Wellcome Trust, the Medical Research Council, Versus Arthritis, the European Union Horizon 2020, the Chronic Disease Research Foundation (CDRF), Zoe Ltd, the National Institute for Health and Care Research (NIHR) Clinical Research Network (CRN) and the Biomedical Research Centre based at Guy's and St Thomas' NHS Foundation Trust in partnership with King's College London.

**Author contributions** M.D.C.N., M.E.H. and R.R. wrote the manuscript. All authors reviewed and edited the manuscript. M.E.H. and R.R. supervised the project. M.P.G., S. Wadge, K.S. and R.R. led sample procurement. M.D.C.N., T.B. and P.A.N. conducted sperm counting. A.R.J.L., P.J.C., K.R., S.V.L., S. Widaa and I.M. contributed to sample sequencing implementation. M.D.C.N. led the analysis of the data with help from A.R.J.L., R.S., F.A., M.H.P., A.C., P.A.N., A.B.-O., R.E.A., M.R.S., P.J.C., I.M., M.E.H. and R.R.

**Competing interests** I.M., M.R.S. and P.J.C. are co-founders, shareholders and consultants for Quotient Therapeutics. R.E.A. is an employee of Quotient Therapeutics. M.E.H. is a co-founder of, consultant to and holds shares in Congenica, a genetics diagnostic company. The remaining authors declare no competing interests.

**Additional information**
**Correspondence and requests for materials** should be addressed to Raheleh Rahbari.

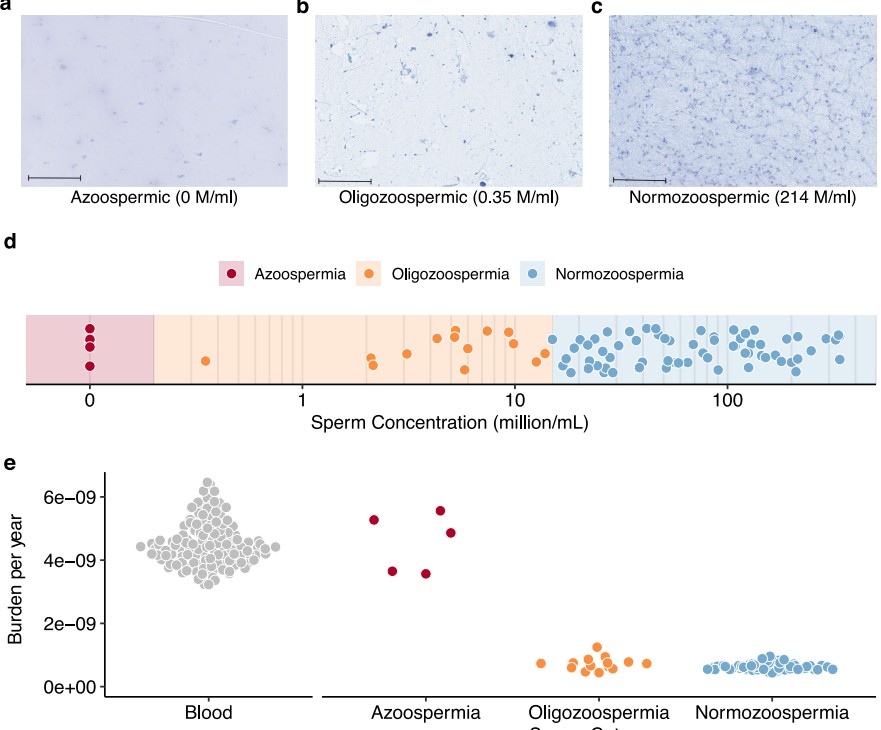

**Extended Data Fig. 1 | Sperm counting. a,b,c,** Slides of Papanicolaou stained semen samples for (**a**) an azoospermic sample where no sperm cells are visible, (**b**) an oligozoospermic sample where a small number of sperm samples are visible and (**c**) a normozoospermic sample where many sperm cells are visible. Sperm concentrations are given for each sample in millions of sperm per ml (M/ml). The black band in the bottom left of each slide photo corresponds to 100 μm. Staining was independently repeated once with similar results. **d,** The distribution of sperm counts on a log scale among semen samples analysed with colour bands indicating the concentration bin of the sample. All samples below 1 million/mL were subsequently excluded. **e,** The distributions of mutation burden per year from blood samples and three categories of sperm samples broken down by sperm concentration.

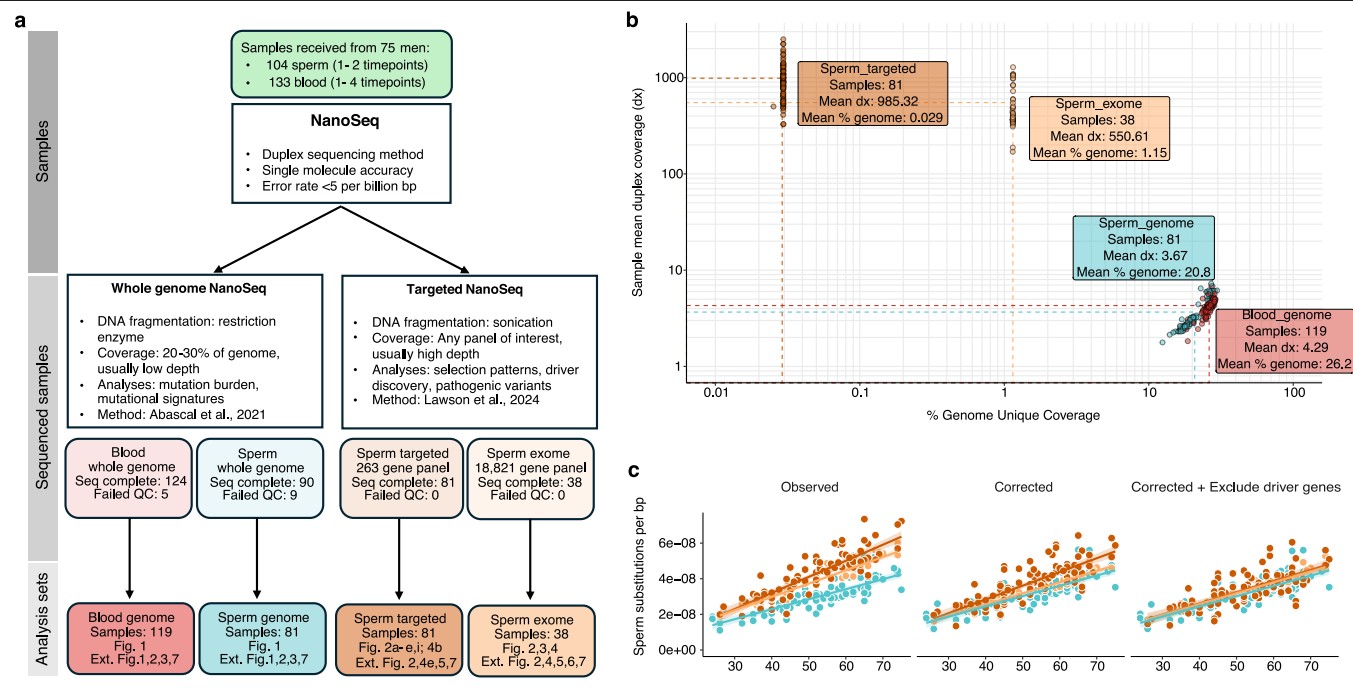

**Extended Data Fig. 2 | Sequencing method and coverage summary.**
**a**, Graphical relationship between NanoSeq methods applied in manuscript. Both sperm and blood samples underwent genome NanoSeq, used for mutation burden and mutational signature analyses. Only sperm samples were used for targeted NanoSeq applied for selection, driver, and pathogenic variant analyses. Targeted NanoSeq is adaptable to different target panels, and we have named the sample sets as "targeted" for the samples using the 263 gene cancer panel and "exome" for samples using the exome wide panel. **b**, Mean duplex coverage (log scale) and percentage of genome covered (log scale) per sample.

Panels summarise the mean duplex coverage (dx) and mean percentage of genome covered per NanoSeq type and tissue. **c**, Mutation burden of targeted (dark orange), exome (yellow), and genome (blue) sperm sequenced samples that were observed without correction (left), corrected for trinucleotide composition of covered base pairs relative to the whole genome (middle) or corrected and masked for mutations and coverage in the 44 genes linked to germline positive selection (right). Models are linear regressions, with the central line showing the model fit and the shaded bands indicating 95% confidence intervals.

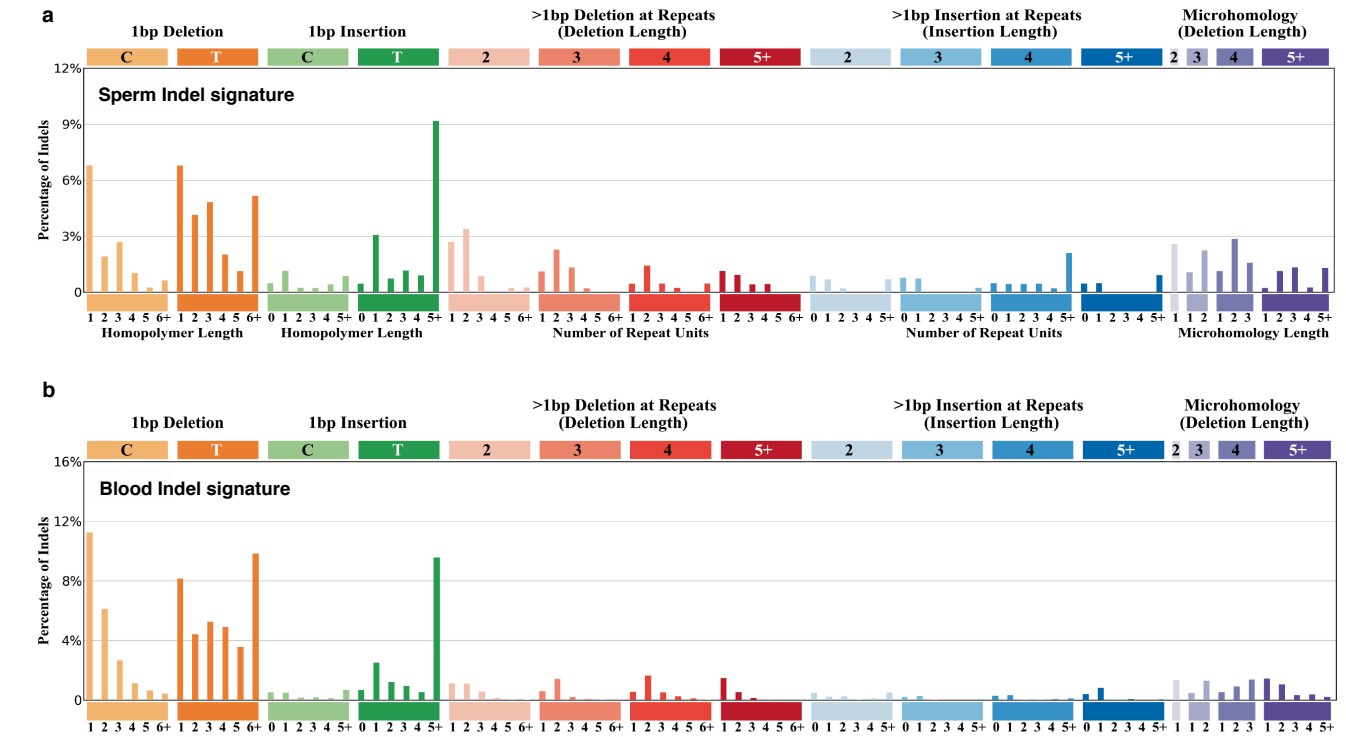

**Extended Data Fig. 3 | Insertion deletion mutation profiles. a,b,** Distribution of indel types observed in whole genome (**a**) sperm and (**b**) blood.

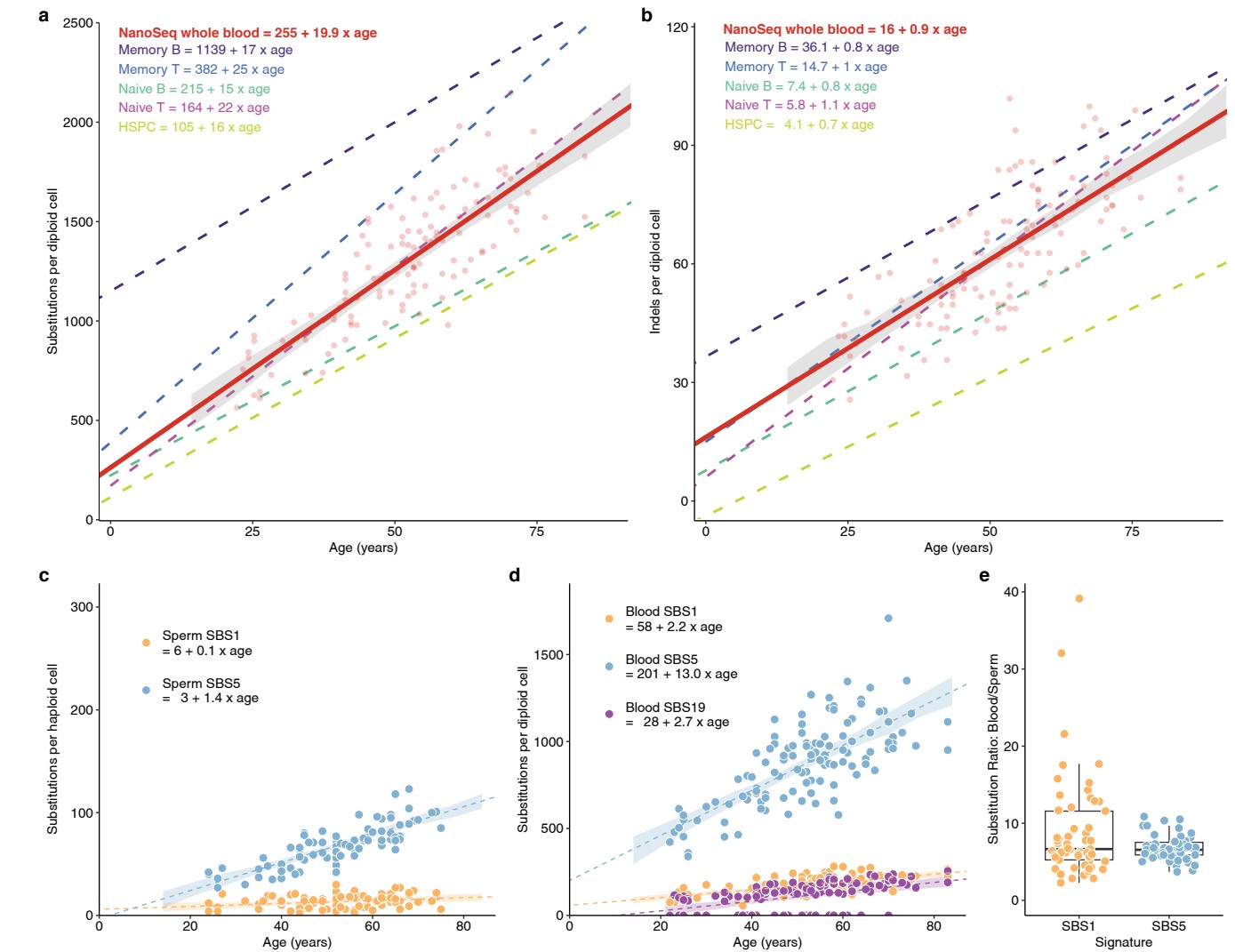

**Extended Data Fig. 4 | Mutation rates relative to blood cell types and split by signatures. a,b,** Substitutions (**a**) and indels (**b**) per diploid cell from blood NanoSeq relative to specific blood cell types[13]. **c,d,** Substitutions per haploid cell for sperm (**c**) and diploid cell for blood (**d**) split by signature contributions of SBS1, SBS5, and SBS19. **a,b,c,d,** Models are linear mixed regressions, with the central line showing the model fit and the shaded bands indicating 95% confidence intervals calculated by parametric bootstrapping. **e,** Ratio of age-corrected blood to sperm substitutions per diploid cell per year for mutations assigned to SBS1 and SBS5. Each dot corresponds to an individual with both a blood and sperm sample and where individuals had multiple timepoints the mean value of all timepoints in that tissue was used. Box plots show the median as center line, the 25th and 75th percentiles as box limits, and whiskers extending to the largest and smallest values within 1.5× the interquartile range from the limits from n = 57 biologically independent samples.

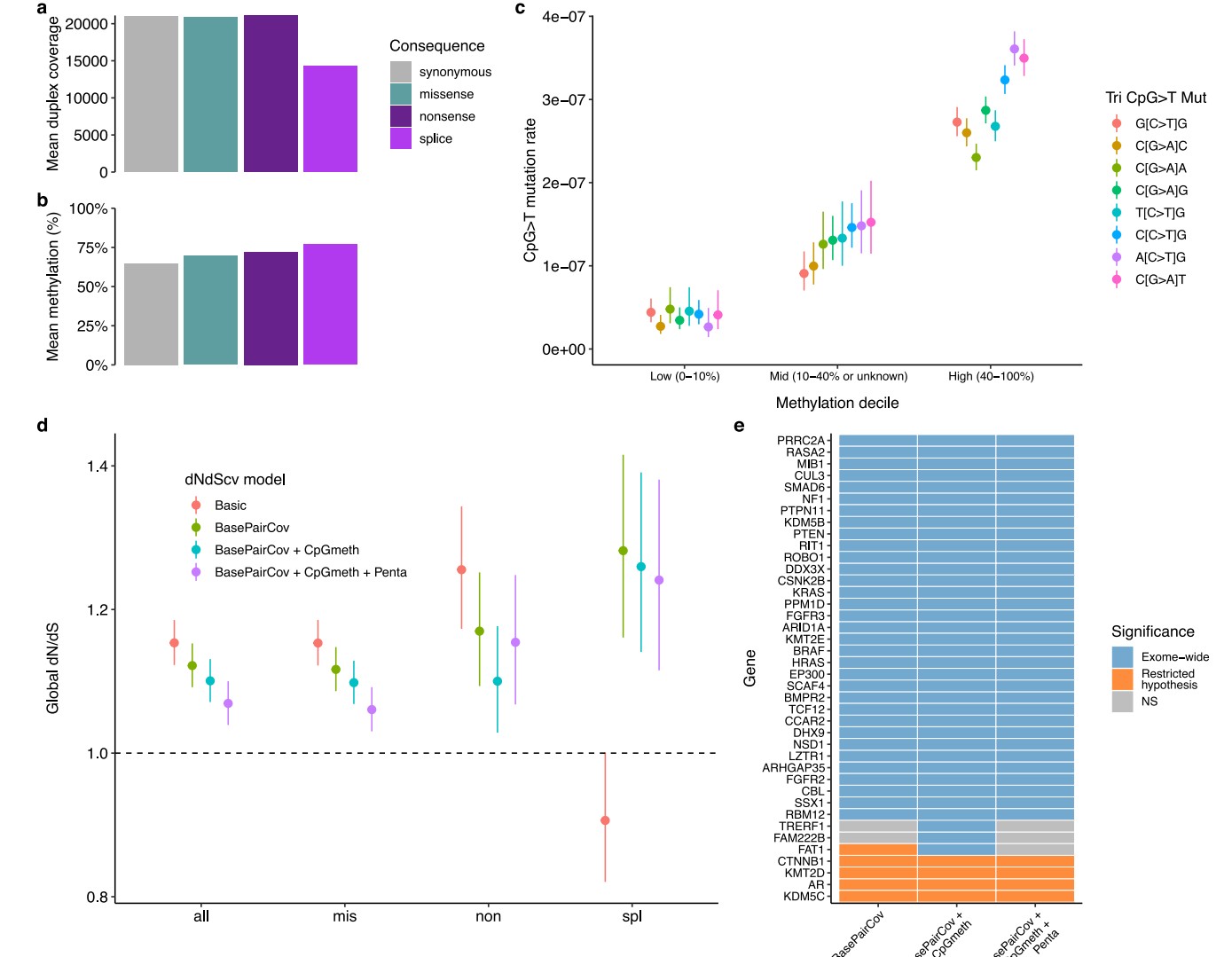

**Extended Data Fig. 5 | Model selection dN/dS. a,b,** Mean duplex coverage (**a**) and methylation percentage (**b**) of all base pairs with exome sequencing coverage split by mutation consequence. **c,** C > T mutation rate point estimates at CpG sites in n = 38 biologically independent exome-sequenced sperm samples split by methylation bin based on percentage methylated from testis bisulfite sequencing. **d,** Comparison of global dN/dS ratio point estimates from mutations of n = 38 exome-sequenced sperm samples using different modifications to the *dNdScv* algorithm. Categories include all nonsynonymous mutations, missense, nonsense or essential splice. The basic model excludes genes with no coverage and uses default parameters. Additional models show the impact of adding corrections for duplex coverage per base pair (BasePairCov), CpG methylation level (CpGmeth), and pentanucleotide context (Penta). **e,** Comparison of per-gene significance in exome-wide (blue) or restricted hypothesis (orange) dN/dS tests using the different models. Genes that did not reach significance in either test are shown in grey. Error bars indicate 95% confidence intervals.

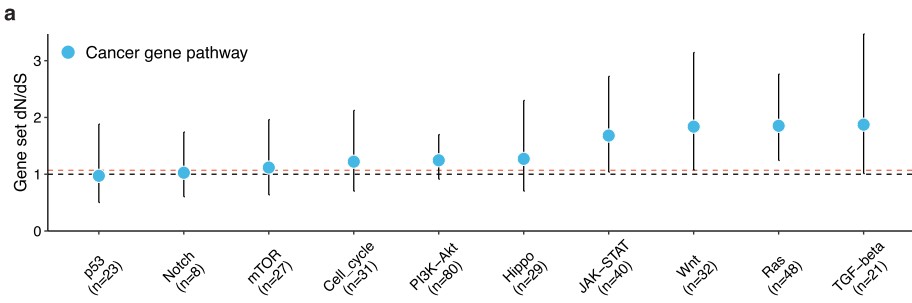

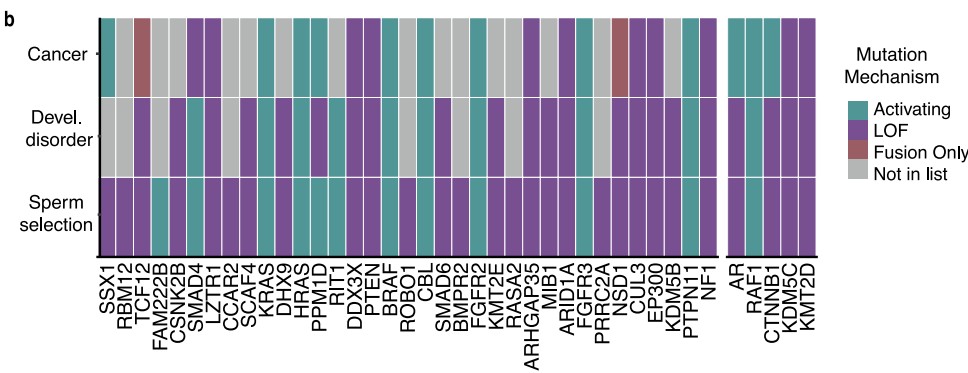

**Extended Data Fig. 6 | Gene mutation mechanisms. a**, dN/dS ratios for sperm SNVs across germline selection genes and cancer gene census genes split by ten canonical cancer pathways, where the dotted black line indicates neutrality and the dotted orange line represents the cohort average across all genes. **b**, The mutation mechanism assigned to each gene based on the mutation pattern in sperm, developmental disorders, and cancer (Methods).

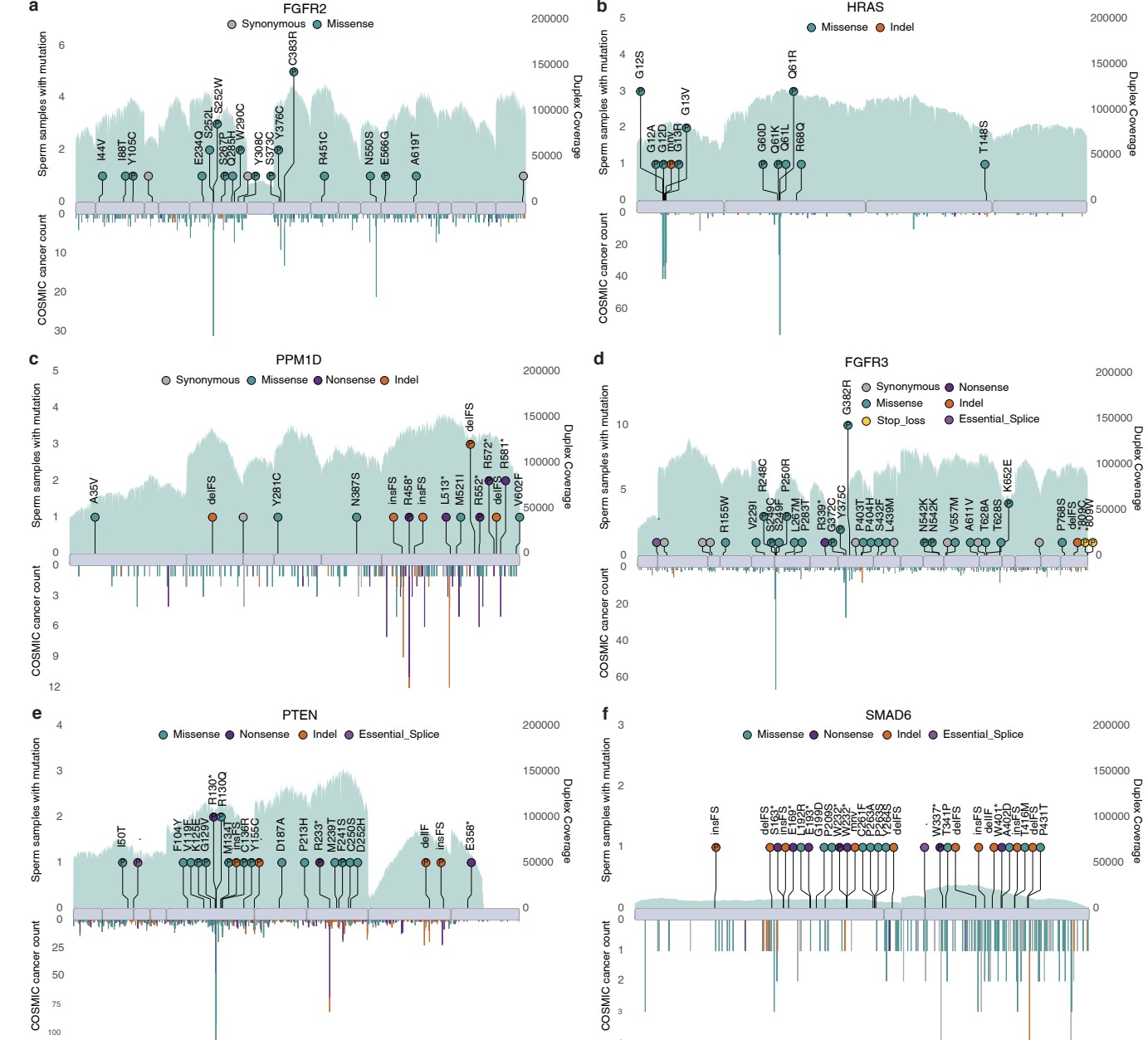

**Extended Data Fig. 7 | Gene mutation patterns. a,b,c,d,e,f,** Observed sperm mutations across the cohort for six illustrative genes where the height of the "lollipop" represents the number of unique samples with a mutation at that location and the colour represents its mutation type. Mutations are labelled with their amino acid consequence for point substitutions or their insertion (ins)/deletion (del) consequence of in frame (IF) or frameshift (FS). A "P" indicates that the variant is classified as pathogenic/likely pathogenic in ClinVar. Exons are shown as purple rectangles and the blue background represents the total duplex coverage across the cohort. Lines below the gene indicate COSMIC somatic mutations in cancer within that gene.

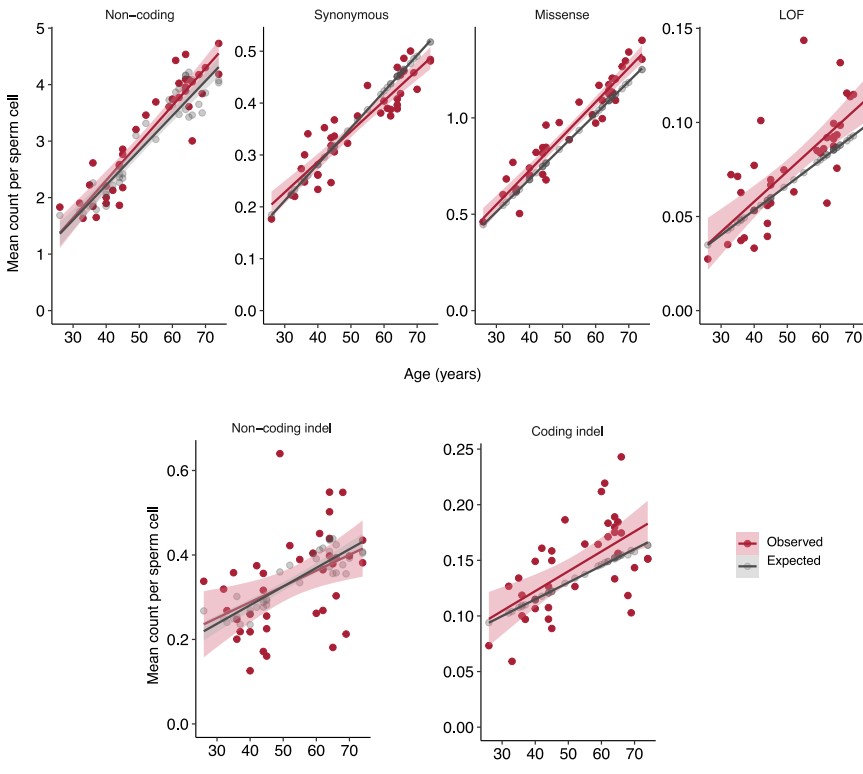

**Extended Data Fig. 8 | Mean variant class count per individual by age.**
The relationship between age and the mean count of SNVs (non-coding, synonymous, missense, and loss-of-function (nonsense or essential splice)) and indels (non-coding indel and coding indel) per sperm cell. The red points represent the observed values for each individual. The grey line represents the expected mutation count per sperm based on the germline mutation rate model. Error bands indicate 95% confidence intervals of linear regressions.

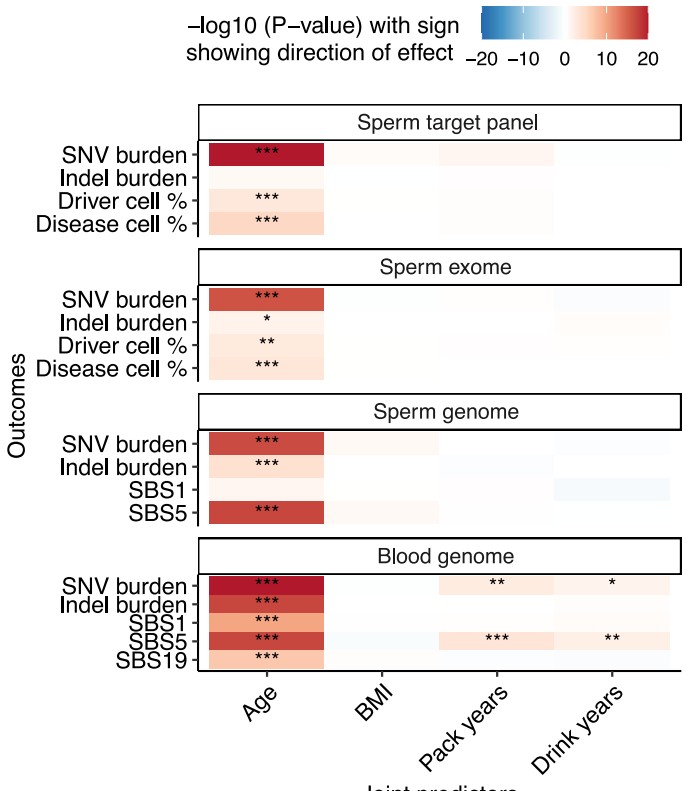

**Extended Data Fig. 9 | Phenotype correlations.** Correlation of cohort phenotypes with mutation outcomes across sequencing datasets. Associations were tested using two-sided generalised linear models (family = gaussian). *P* values were adjusted for multiple comparisons using the false discovery rate (FDR) method. Asterisks indicate FDR-corrected *P* value ranges: (*P* value > 0.01 to <0.05, **P* value > 0.001 to <0.01, ***P* value < 0.001).

**Extended Data Table 1 | Significant SNV hotspots from dN/dS exome-wide and restricted hypothesis tests (RHT)**

| chr | pos | ref | mut | gene | aachange | impact | obs | exp | dnds_ratio | qval_exome | qval_RHT |
|---|---|---|---|---|---|---|---|---|---|---|---|
| 4 | 1806119 | G | A | FGFR3 | G382R | Missense | 10 | 0.007750 | 1290 | 2.69e-06 | 2.33e-11 |
| 10 | 123274774 | A | G | FGFR2 | C383R | Missense | 5 | 0.000803 | 6226 | 3.33e-05 | 2.88e-10 |
| 4 | 1807889 | A | G | FGFR3 | K652E | Missense | 4 | 0.000225 | 17816 | 3.33e-05 | 2.88e-10 |
| 12 | 112926909 | A | G | PTPN11 | Q510R | Missense | 5 | 0.000925 | 5406 | 4.46e-05 | NA |
| 12 | 112888168 | T | G | PTPN11 | Y62D | Missense | 4 | 0.000303 | 13198 | 6.62e-05 | NA |
| 18 | 48604664 | C | T | SMAD4 | R496C | Missense | 9 | 0.011300 | 793 | 1.81e-04 | 2.20e-09 |
| 12 | 112888172 | A | G | PTPN11 | Y63C | Missense | 6 | 0.003010 | 1994 | 1.81e-04 | 2.20e-09 |
| 18 | 48604676 | A | G | SMAD4 | I500V | Missense | 5 | 0.002230 | 2244 | 1.73e-03 | 2.00e-08 |
| 17 | 27086078 | C | T | FAM222B | R300H | Missense | 5 | 0.002320 | 2157 | 1.87e-03 | NA |
| 22 | 41572254 | T | G | EP300 | F1595V | Missense | 3 | 0.000176 | 17044 | 2.90e-03 | 3.58e-08 |
| 12 | 112926890 | A | G | PTPN11 | M504V | Missense | 4 | 0.000963 | 4154 | 2.90e-03 | 3.58e-08 |
| 12 | 112926887 | G | A | PTPN11 | G503R | Missense | 4 | 0.000975 | 4101 | 2.90e-03 | NA |
| 1 | 155874263 | T | C | RIT1 | M107V | Missense | 5 | 0.003180 | 1572 | 6.09e-03 | NA |
| 4 | 1803571 | C | G | FGFR3 | P250R | Missense | 3 | 0.000337 | 8912 | 1.64e-02 | NA |
| 1 | 155874285 | A | C | RIT1 | F99L | Missense | 3 | 0.000349 | 8600 | 1.70e-02 | NA |
| 10 | 123279677 | G | C | FGFR2 | S252W | Missense | 3 | 0.000491 | 6106 | 4.45e-02 | 6.85e-07 |
| 18 | 19426998 | C | T | MIB1 | R769* | Nonsense | 4 | 0.002190 | 1825 | 5.05e-02 | NA |
| 12 | 112926270 | C | T | PTPN11 | T468M | Missense | 4 | 0.003070 | 1301 | 1.80e-01 | 2.81e-06 |
| 12 | 112888199 | C | T | PTPN11 | A72V | Missense | 3 | 0.000861 | 3484 | 2.00e-01 | 3.00e-06 |
| 12 | 112926851 | C | T | PTPN11 | P491S | Missense | 3 | 0.000936 | 3205 | 2.44e-01 | 3.50e-06 |
| 11 | 533874 | T | C | HRAS | Q61R | Missense | 3 | 0.000959 | 3127 | 2.50e-01 | 3.50e-06 |
| 3 | 12645699 | G | A | RAF1 | S257L | Missense | 4 | 0.012700 | 314 | 1.00e+00 | 4.61e-04 |
| 4 | 1803564 | C | T | FGFR3 | R248C | Missense | 3 | 0.005820 | 516 | 1.00e+00 | 6.16e-04 |
| 11 | 534289 | C | T | HRAS | G12S | Missense | 3 | 0.010100 | 297 | 1.00e+00 | 2.79e-03 |
| 17 | 29654736 | C | T | NF1 | R1830C | Missense | 3 | 0.013200 | 228 | 1.00e+00 | 5.53e-03 |

# Reporting Summary

## Statistics

For all statistical analyses, confirm that the following items are present in the figure legend, table legend, main text, or Methods section.

| n/a | Confirmed | |
|---|---|---|
| ☐ | ☒ | The exact sample size (*n*) for each experimental group/condition, given as a discrete number and unit of measurement |
| ☐ | ☒ | A statement on whether measurements were taken from distinct samples or whether the same sample was measured repeatedly |
| ☐ | ☒ | The statistical test(s) used AND whether they are one- or two-sided<br>*Only common tests should be described solely by name; describe more complex techniques in the Methods section.* |
| ☐ | ☒ | A description of all covariates tested |
| ☐ | ☒ | A description of any assumptions or corrections, such as tests of normality and adjustment for multiple comparisons |
| ☐ | ☒ | A full description of the statistical parameters including central tendency (e.g. means) or other basic estimates (e.g. regression coefficient) AND variation (e.g. standard deviation) or associated estimates of uncertainty (e.g. confidence intervals) |
| ☐ | ☒ | For null hypothesis testing, the test statistic (e.g. *F*, *t*, *r*) with confidence intervals, effect sizes, degrees of freedom and *P* value noted<br>*Give P values as exact values whenever suitable.* |
| ☒ | ☐ | For Bayesian analysis, information on the choice of priors and Markov chain Monte Carlo settings |
| ☒ | ☐ | For hierarchical and complex designs, identification of the appropriate level for tests and full reporting of outcomes |
| ☒ | ☐ | Estimates of effect sizes (e.g. Cohen's *d*, Pearson's *r*), indicating how they were calculated |

*Our web collection on statistics for biologists contains articles on many of the points above.*

## Software and code

Policy information about availability of computer code

| Data collection | None. |
|---|---|
| Data analysis | NanoSeq variant calling pipeline (https://github.com/cancerit/NanoSeq; v3.3 was used to process duplex sequencing data. R (v4.3.1) was used for statistical analysis and visualization. Code modified from trackViewer (R/Bioconductor package v1.38.0) was used to generate gene mutation "lollipop" plots. lme4 (v1.1-33) was used for linear mixed-effects models. ggplot2 (v3.4.4) was used for plotting. dNdScv (https://github.com/im3sanger/dndscv; version as of commit on Sep 29, 2023) was used for selection analysis. HDP (https://github.com/nicolaroberts/hdp) was used for de novo mutational signature extraction. SigProfilerAssignment (v0.0.27; https://github.com/AlexandrovLab/SigProfilerAssignment) was used for COSMIC signature fitting.<br>All custom analysis code and scripts are available at https://github.com/mattnev17/spermPositiveSelectionManuscript. |

For manuscripts utilizing custom algorithms or software that are central to the research but not yet described in published literature, software must be made available to editors and reviewers. We strongly encourage code deposition in a community repository (e.g. GitHub). See the Nature Portfolio guidelines for submitting code & software for further information.

## Data

Policy information about <u>availability of data</u>

All manuscripts must include a <u>data availability statement</u>. This statement should provide the following information, where applicable:

- Accession codes, unique identifiers, or web links for publicly available datasets
- A description of any restrictions on data availability
- For clinical datasets or third party data, please ensure that the statement adheres to our <u>policy</u>

Raw sequencing data are available on the European Genome-Phenome Archive under accession number X. All non TwinsUK files necessary to recreate results are available at https://github.com/mattnev17/spermPositiveSelectionManuscript. Additional TwinsUK individual-level data are not permitted to be publicly shared or deposited due to the original consent given at the time of data collection, where access to these data is subject to governance oversight. All data access requests are overseen by the TwinsUK Resource Executive Committee (TREC). Requests will be reviewed within 4–6 weeks. For information on access to these genotype and phenotype data and how to apply, see https://twinsuk.ac.uk/researchers/access-data-and-samples/request-access/.
This study also made use of the following publicly available resources: the Developmental Disorder Genotype–Phenotype Database (DDG2P) at https://www.deciphergenomics.org/ddd/ddgenes, Online Mendelian Inheritance in Man (OMIM) at https://omim.org, the COSMIC Cancer Gene Census at https://cancer.sanger.ac.uk/census, gnomAD v2.1.1 at https://gnomad.broadinstitute.org, ClinVar at https://www.ncbi.nlm.nih.gov/clinvar/, and bisulfite methylation data from the ENCODE Project (testis samples ENCFF638QVP and ENCFF715DMX) available at https://www.encodeproject.org.

## Research involving human participants, their data, or biological material

Policy information about studies with <u>human participants or human data</u>. See also policy information about <u>sex, gender (identity/presentation), and sexual orientation</u> and <u>race, ethnicity and racism</u>.

| | |
|---|---|
| Reporting on sex and gender | All individuals self reported as male and provided a sperm sample. |
| Reporting on race, ethnicity, or other socially relevant groupings | All individuals that provided ethnicity information self-reported as "white". |
| Population characteristics | Sperm samples were collected at ages 24-75 years. Blood samples were collected at ages 22-83. Sample count, timepoint, and twin relationships are summarized in Supplementary Table 1. |
| Recruitment | Participants were recruited through the TwinsUK registry, a volunteer cohort of adult twins in the UK. Inclusion in this study was based on availability/willingness to donate high-quality semen and blood samples from male participants, as well as consent for genetic research. As a volunteer-based cohort, there is potential for self-selection bias, including overrepresentation of individuals who are more health-conscious or engaged with research. While this may limit generalizability to the broader population, the biological processes under study—mutation accumulation and selection in spermatogenesis—are unlikely to be strongly influenced by these factors. We also note that no individuals in the cohort had known genetic disorders or exposures (e.g. chemotherapy) likely to affect germline mutation patterns. |
| Ethics oversight | This study was carried out under TwinsUK BioBank ethics, approved by North West – Liverpool Central Research Ethics Committee (REC reference 19/NW/0187), IRAS ID 258513 and earlier approvals granted to TwinsUK by the St Thomas' Hospital Research Ethics Committee, later London – Westminster Research Ethics Committee (REC reference EC04/015). |

Note that full information on the approval of the study protocol must also be provided in the manuscript.

# Field-specific reporting

Please select the one below that is the best fit for your research. If you are not sure, read the appropriate sections before making your selection.

☒ Life sciences ☐ Behavioural & social sciences ☐ Ecological, evolutionary & environmental sciences

For a reference copy of the document with all sections, see nature.com/documents/nr-reporting-summary-flat.pdf

# Life sciences study design

All studies must disclose on these points even when the disclosure is negative.

| | |
|---|---|
| Sample size | The number of samples was determined by availability from the TwinsUK cohort. |
| Data exclusions | Some samples were excluded based on failed sequencing (Supplementary Table 1). Six sperm samples were excluded based on sperm count <1M/ml, which was not predetermined but based off of analyses in Supplementary Note 1. Three blood samples were excluded based off of a predetermined contamination threshold of verifyBamID alpha value above 0.005 (Abascal et al, 2021). Three sperm samples were excluded based off of a verifyBamID alpha value above 0.002, which was determined by analyses in the paper showing sperm needed a more stringent threshold than blood. |
| Replication | The mutation burdens described in sperm and blood were verified by comparing to well known estimates of these values from previous |

| | |
|---|---|
| Replication | publications. Analyses of positive selection replicated 9 of 13 known genes that drive this effect but otherwise describes the landscape of mutations using all available sample material. |
| Randomization | Randomization was not relevant to this study because it is an observational analysis of naturally occurring mutations in human sperm and blood samples. No interventions or group assignments were made, and sample comparisons (e.g., by age) reflect inherent biological variation rather than experimental manipulation. |
| Blinding | Blinding was not relevant to this study because it involved computational analysis of sequencing data without subjective measurements or investigator-assigned interventions. All variant calling, mutation burden estimation, and statistical analyses were performed using predefined pipelines or automated methods, minimizing risk of bias. |

# Reporting for specific materials, systems and methods

We require information from authors about some types of materials, experimental systems and methods used in many studies. Here, indicate whether each material, system or method listed is relevant to your study. If you are not sure if a list item applies to your research, read the appropriate section before selecting a response.

## Materials & experimental systems

| n/a | Involved in the study |
|---|---|
| ☒ ☐ | Antibodies |
| ☒ ☐ | Eukaryotic cell lines |
| ☒ ☐ | Palaeontology and archaeology |
| ☒ ☐ | Animals and other organisms |
| ☒ ☐ | Clinical data |
| ☒ ☐ | Dual use research of concern |
| ☒ ☐ | Plants |

## Methods

| n/a | Involved in the study |
|---|---|
| ☒ ☐ | ChIP-seq |
| ☒ ☐ | Flow cytometry |
| ☒ ☐ | MRI-based neuroimaging |

## Plants

| | |
|---|---|
| Seed stocks | *Report on the source of all seed stocks or other plant material used. If applicable, state the seed stock centre and catalogue number. If plant specimens were collected from the field, describe the collection location, date and sampling procedures.* |
| Novel plant genotypes | *Describe the methods by which all novel plant genotypes were produced. This includes those generated by transgenic approaches, gene editing, chemical/radiation-based mutagenesis and hybridization. For transgenic lines, describe the transformation method, the number of independent lines analyzed and the generation upon which experiments were performed. For gene-edited lines, describe the editor used, the endogenous sequence targeted for editing, the targeting guide RNA sequence (if applicable) and how the editor was applied.* |
| Authentication | *Describe any authentication procedures for each seed stock used or novel genotype generated. Describe any experiments used to assess the effect of a mutation and, where applicable, how potential secondary effects (e.g. second site T-DNA insertions, mosiacism, off-target gene editing) were examined.* |

