## [Peer Review File · Nature]

Sperm sequencing reveals extensive positive selection in the male germline

Corresponding Author: Dr Raheleh Rahbari

Version 0:

Reviewer comments:

Referee #1

(Remarks to the Author)

The manuscript by Neville et al. describes an interesting study on the use of duplex sequencing in sperm samples to study positive selection of certain mutated genes in the male germline. The most important novel observation is that 40 genes (of which 31 novel) were found to be under significant positive selection in the male germline. Overall, the work is of interest, but there are some important limitations to the study and the results could be presented better.

Limitations: The study is done in a single cohort using a single technology. Independent replication in another cohort would be critical to boost confidence in the results described, preferably also using another technology.

Unclear: It would be useful to describe early in the results section how mutations were called in sperm samples and how comparison to matched blood samples helped in distinguishing inherited from new mutations. In line 76 the authors mention 6,653 SNVs detected in sperm, I assume these are the SNVs not present in blood, but this is not clearly described in the manuscript.

Overall, the study design is complex and involves the analysis of different sample numbers for different approaches, including three different targeted gene panels as well as exome sequencing. It is very difficult to understand what was done, what the concordance was between the different approaches and understand the reliability of the approaches. This is further complicated by the identification of DNA contamination which was acknowledged by the authors. While the authors claim they were able to in silico decontaminate their results, it does make one wonder about the overall robustness of the data (hence the request for independent replication, see above).

Details: Unclearities very much relate to different sample numbers used and referred to throughout the manuscript. After reading the manuscript a few times I am still not clear on some numbers referred to in the manuscript. The results section (line 64) refers to 81 semen samples for which whole genome Nanoseq was done, whereas supplementary note 1 it seems like sperm counting was done in 85 samples of which 6 were excluded, so 79 in total? Also, line 67 talks about that these sperm samples and matched blood comes from 63 men, whereas Suppl. Table 1 mentions 62 men, and only 57 passed analysis.

(Remarks on code availability)

Referee #2

(Remarks to the Author)

This manuscript by Neville et al detected de novo mutations in 81 sperm samples from 63 men in the TwinsUK cohort by using a high-accuracy single-molecular duplex sequencing technique (NanoSeq). Based on excess non-synonymous mutations relative to synonymous mutations, the authors detected evidence of positive selection across the exome as well as on ~40 genes. As many of these genes are known to be associated with cancer predisposition and developmental disorders, the authors estimated that disease-causing mutations are enriched in sperm by 2-3 fold, driven by the competitive advantages that they confer during spermatogenesis.

Overall, the paper is well written and showcases the combined strengths of several groups in various fields of mutation study (e.g., cancer genomics, germline mutagenesis, mutational basis of developmental disorders, and technology and computational method development). As the baseline (neutral) mutation model is central to the selection test, my only major

concern is the sensitivity of the results to the mutation model, and I also have some questions regarding the mutation model itself. Additionally, I noticed that the authors planned to make the raw sequencing data available on EGA (with accession number pending) but not the sperm mutation calls. Since these sperm mutations are technically not part of the individual genotypes, could the authors release these mutation calls in the supplementary tables? Unless this data release is prohibited by consent or other regulations, I think these data should be released in accordance with Nature's policy.

Major comments:

(1) Inference of selection

a. As Martincorena et al (2017) demonstrated, the use of imprecise mutation models can lead to systematic bias in estimates of dN/dS and incorrect inference of selection. In this manuscript, the authors went to great lengths to improve the mutation model, including modeling base-specific sequencing coverage, methylation level and 5-mer sequence context. Extended Figure 6d showed that each of these refinements of the model had non-negligible impact on the estimate of dN/dS ratios. This raises a natural question: is the current model sufficiently precise that further refinements (such as 7-mer context or finer groupings of methylation levels) would not lead to further downward adjustment in the dN/dS estimates? Is there empirical data (i.e., adding additional features do not change the results) or semi-quantitative argument to address this possibility?

b. It is surprising that a significant signal of selection could be detected, given that the vast majority of mutations (99.5%) were seen in only a single duplex molecule. Such mutations in the sperm pool are analogous to singletons in polymorphism datasets. Extremely rare variants are expected to deviate little from the mutation input as there is limited opportunity for selection to influence their trajectories, and I suspect that the dN/dS ratio for extremely rare polymorphisms would not deviate significantly from neutral expectation, despite the large amount of data. Is the additional power due to the intermediate coverage per site (which is analogous to sample size in polymorphism dataset)? Or is it due to the special clonal structure of seminiferous tubules? Can the authors perform power simulations to demonstrate that it is feasible to detect signal of selection in rare variants?

(2) Details of dNdScv algorithm and its application to data

Although Martincorena et al (2017) clearly explained and benchmarked the dNdCcv algorithm, it would still be helpful if the authors could briefly outline the model for inferring the background mutation rates, annotating variant effect, and clarify exactly how the model is applied to the data.

a. It is unclear how the effect of a variant in a gene is determined. Specifically, a variant may have different effects on different splicing isoforms. How are these differences reconciled and integrated into a single effect?

b. The application of the dNdScv algorithm was insufficiently described. For example, for results shown in Fig 2f, it is unclear whether three mutation models were inferred separately based on mutations observed in each age group, or if only one mutation model was inferred based on all sperm mutations identified, with the same model being applied to calculate dN/dS ratios of observed mutations in the three age groups. It has been shown that the mutation spectrum of germline mutations significantly varies with age at reproduction for both parents (for example, CpG>TpG mutations are more enriched in DNMs passed down from older fathers), which suggests it may be necessary to infer the background mutation model for each age group. Related to this, do the authors observe correlation between the fraction of SBS1 mutations with age?

c. Similarly, it is unclear whether gene expression level was explicitly modeled as a feature and whether the same or different mutation models were used for estimating dN/dS for mutations in genes of different expression levels. If only one mutation model based on all mutations was inferred, the high dN/dS ratio of highly expressed genes (Fig 2g) may reflect deviation in the mutation profile of these genes from the exome average due to transcription-associated molecular processes.

Minor points:

Line 24 (abstract): the sentence "while four genes that exhibit..." is grammatically problematic.

Line 33: "replicating (tissues)" should be "dividing" or "proliferating".

Line 66: the authors mentioned 9 pairs of monozygotic twins in the main text, but 8 pairs in the supplemental note 2. Why the difference?

Lines 323-324: The phrase "on known driver genes" should be added after "germline positive selection". Otherwise, it seems to imply other non-selection mechanism could drive the higher than expected rate of disease-causing mutations.

Line 691: Should the word "multiplying" be "dividing"? Alternatively, the ratio should be "the total genome size to the sample's callable genome".

(Remarks on code availability)

Referee #3

(Remarks to the Author)

The manuscript presents a compelling and technically well-executed study that provides significant implications into reproductive biology and the dynamics of germline mutations, particularly in the context of paternal age and disease genetic risks. Utilising the duplex sequencing method NanoSeq, the authors identify positively selected genes in spermatogonia by sequencing bulk sperm (and blood as a "control"), identifying 40 genes under positive selection in the male germline linked to developmental disorders or cancer predisposition disorders in children. Their analyses indicate that the dominant selection force on germline mutations is positive selection during spermatogenesis but negative selection between generations. The findings presented, and the far reaching implications of 3-5% of sperm from older individuals harboring pathogenic mutations, represent a very significant advance and this is a well-written and timely study. NanoSeq appears to be the right method for the study presented; a relevant update to NanoSeq (capture-based) is employed and the authors

convincingly show how this enhances the power for identifying mutations positively selected in spermatogonia. Still there are several areas where the manuscript could be strengthened to enhance clarity, depth, and rigor.

o Duplex sequencing coverage seems adequate but somewhat on the low side. It would be helpful to detail coverage statistics across repeat classes and discuss limitations, particularly in segmentally duplicated regions and STRs. The authors should discuss, and quantify, the likely under-representation of reads in repeat-rich regions affecting variant calling. I wonder, would the authors even be powered to uncover evidence of (recurrent) de novo STRs? I would not necessarily expect the authors to solve this issue, but it should be spelled out clearly in the discussion.

o The manuscript should address the potential biases introduced by the choice of restriction enzyme in the NanoSeq protocol. A discussion of whether different enzymes might yield varying results and how this could affect conclusions is warranted.

o The capture based approach to yield higher duplex coverage is a well chosen experimental setup, allowing the authors to sensitively detect positive selection and revealing an increase in the dN/dS ratio with age in genes expressed during spermatogenesis. The manuscript would need to provide a detailed description of this deep whole-exome/targeted NanoSeq method

o The reported increase in the dN/dS ratio with age is an interesting finding (Line 153). However, it is unclear if this trend is statistically significant. Could the authors provide additional details on the statistical tests performed and include a clearer presentation of significance in Figure 2f?

o Including a gene not expressed in spermatogonial cells as a negative control in Figure 2abc would strengthen the evidence for selective pressure in genes expressed during spermatogenesis.

o While age is a dominant factor influencing mutational burden, the potential effects of BMI, smoking, and alcohol consumption on mutational signatures and dN/dS ratios should be explored in greater detail. Were there subtle trends, even if not statistically significant, that might warrant further investigation in larger cohorts?

o The manuscript does not discuss whether structural variants or copy-neutral loss-of-heterozygosity (LOH) events were observed in the dataset. This could be an interesting avenue to explore, given the implications for germline mutations.

o The findings should be explicitly linked to the 'selfish spermatogonial selection' hypothesis in the Discussion. How do these results refine or expand our understanding of the theory?

o I would welcome a more extensive discussion of the bias against tumor suppressor mutations, as opposed to activating missense mutations, amongst positively selected genes. I could imagine that inactivated tumor suppressors may have a stronger effect across distinct cell types and tissues and thus more often result in embryonic viability, but suppose there may be other explanations too

o A more detailed discussion of potential clinical implications of the findings, particularly for counseling older prospective fathers, would enhance the manuscript's impact.

Issues of minor concern:

- The sentence in the abstract: "Most positively selected genes are associated with developmental or cancer predisposition disorders in children, while four genes that exhibit elevated frequencies of protein-truncating variants in healthy populations." is awkwardly phrased. Please improve clarity.

- The statement that "spermatogonial stem cells are the only replicating cells with the potential to transmit mutations to offspring" seems incorrect to me. Recent work (e.g., Porubsky et al., 2024; <https://www.biorxiv.org/content/10.1101/2024.08.05.606142v1>), highlights that transmitted mutations may also arise post-zygotically in replicating cells.

- Fig. 2abc could benefit from clearer labeling, and Fig. 3c should indicate whether observed age trends are statistically significant.

- Are sperm counts particularly linked with the number of mutations acquired in sperm?

(Remarks on code availability)

Version 1:

Reviewer comments:

Referee #1

(Remarks to the Author)

The authors have responded in detail to all of my comments, improved the manuscript and convinced me of the quality, reliability and importance of the work. I therefore have no further questions.

(Remarks on code availability)

Referee #2

(Remarks to the Author)

The authors did an excellent job putting together the response to reviewers, which adequately addressed all my major questions. Below are some minor points that I hope they could consider incorporating in the manuscript:

1. Mutation calling accuracy (previously raised by reviewer1)

Although Fig 1 and Rebuttal Fig 1 clearly show that the mutation burden, as a function of age/sample, are highly concordant across cohorts/methods, these results do not directly speak to the accuracy of individual mutation calls, as it is possible that false positives are roughly balanced by a similar number of false negatives, yet the rates of both are substantial. Since detection of positive selection is reliant on the exact mutation calls rather than the mutation burden per sample, it is important to demonstrate that the individual mutation calls are highly reliable. It will be helpful to include an estimate of the false discovery rate of NanoSeq (I understand FNR is more challenging to estimate).

2. Sensitivity to mutation model

I understand and agree with the authors' argument that training mutation model with longer sequence context and finer methylation bins are infeasible given current data and the resulting changes in the results may be marginal. However, a general point I wanted to make is that the positive selection test relies on the null mutation model, and inference of the mutation model always makes certain assumptions and is still rapidly improving. Therefore, the specific dN/dS values (exome-wide or for sets of genes) may be subject to revision with further improved mutation models that become available in the future (although I have little doubt that most of the candidate genes are robust to the baseline mutation model). I hope the authors can add a brief comment on this in the description of results, or in the Discussion.

3. In lines 40-41, "replicating cells" can perhaps be better phrased as "proliferating cells" or "dividing cells". It will also be helpful to add a qualifier "in the adult body" to these sentences, which will address the comment of reviewer 3 on postzygotic mutations (which I believe reviewer 3 meant for early embryonic mutations in the parent; these mutations technically arise in dividing cells and can be indeed transmitted to offspring).

(Remarks on code availability)

Referee #3

(Remarks to the Author)

The authors have done a very nice job in refining their manuscript based on the comments I provided

(Remarks on code availability)

Response to Referees Letter

We are grateful for the time and thoughtful feedback provided by the referees on our manuscript, “*Sperm sequencing reveals extensive positive selection in the male germline*”. Their comments have been invaluable in refining our work, and we are pleased to submit a revised version along with a detailed point by point response addressing their concerns.

The original reviewers' comments and questions are in blue, while our responses are in black. Line numbers of changes to the Manuscript and Supplement are noted throughout, which should also be visible as tracked changes in the source .docx files in the attached documents.

Referee #1

R1.1 The manuscript by Neville et al. describes an interesting study on the use of duplex sequencing in sperm samples to study positive selection of certain mutated genes in the male germline. The most important novel observation is that 40 genes (of which 31 novel) were found to be under significant positive selection in the male germline. Overall, the work is of interest, but there are some important limitations to the study and the results could be presented better.

Limitations: The study is done in a single cohort using a single technology. Independent replication in another cohort would be critical to boost confidence in the results described, preferably also using another technology.

We appreciate the reviewer's interest in our study and their useful comments, which helped us to enhance the clarity of our manuscript. To give more confidence to the reviewer, we outline below the ways in which we believe the applied technology has been validated for this purpose and to what degree the results are independently replicated including a newly released preprint led by other researchers, using an orthogonal approach which may directly address the reviewer's request. Below, we consider validation and replication at two levels: the accuracy of the sperm NanoSeq method, and the inference of genes under positive germline selection.

1. The accuracy of NanoSeq and mutation calling in sperm: The accuracy of mutation calling is the most important quality metric of the NanoSeq technology and is the foundation on which all other results, such as detection of positive selection are based.
 - a. Validation: The NanoSeq technology has been carefully validated for accurate mutation calling in multiple cell types including blood, cord blood and specifically in sperm. This was described in depth, in Abascal et al. 2021 (**Rebuttal Fig. 1a**) and mutation burden for targeted/exome NanoSeq was similarly validated and described in Lawson et al. 2024 (**Rebuttal Fig. 1b**).
 - b. Independent replication: In our study we compared observed genome NanoSeq SNV mutation rates, indel mutation rates and mutational signatures from sperm and blood with independent estimates from standard WGS approaches. Firstly, we compared our estimates from sperm NanoSeq with

estimates derived from parent-offspring trio sequencing studies and spermatogonia in testis, each sequenced with standard WGS. We showed high concordance in both the overall mutational burden and mutational signatures between these methods (**Fig. 1a,b**). Reassuringly, our mutation burden estimates from blood samples in this cohort (**Fig. 1d**) also align with findings from Mitchell et al. 2022, which used a single-cell-derived blood colony approach (**Extended Data Fig. 3a,b**). Consistent with the findings from Lawson et al. 2024, we also validated that the mutation burdens we observe in targeted/exome sperm NanoSeq are consistent with those from genome sperm NanoSeq (**Extended Data Fig. 2c**).

Furthermore, sperm mutation rate was independently replicated with a different duplex sequencing method, HiDEF-seq (Liu et al. 2024) (**Rebuttal Fig. 1c**).

Together, these results show that mutation properties from the sperm NanoSeq methods applied are extensively validated and replicated across different methods and cohorts, demonstrating accurate sequencing in sperm.

Rebuttal Fig. 1 | Validation of NanoSeq mutation calling accuracy

a, Sperm NanoSeq SNV burden estimates compared to trio DNM estimates, adapted from Abscal et al 2021. **b**, Consistent mutation burden estimates for cord-blood granulocytes from sonication (Sonic) or enzymatic (Enz) fragmentation protocols, used for targeted and genome NanoSeq respectively. Adapted from Lawson et al 2024. **c**, HiDEF-seq, NanoSeq, and trio DNM estimates for mutation burden vs age in sperm.

2. Genes under positive selection: We secondly address the validation and replication of what the reviewer highlights as our key finding: the 40 genes under positive selection in sperm. The concept of genes under positive selection in the germline was first established through sperm allele-specific assays over 20 years ago (Goriely et al. 2003). Since then, this concept has been repeatedly validated through numerous replications, functional studies, and cross validations in sperm, testis, and birth rates

analyses (e.g. Choi et al. 2012; Maher et al. 2014, 2016; Eboreime et al. 2022; Wood and Goriely 2022). Additionally, it aligns with the now well-established somatic tissue selection observed across all proliferating tissues that have been studied to date (e.g. Martincorena et al. 2015, 2018; Brunner et al. 2019; Lee-Six et al. 2019; Moore et al. 2020; Lawson et al. 2020; Mitchell et al. 2022; Bernstein et al. 2024). We hope the reviewer will therefore agree that the existence of driver mutations under positive selection in the male germline is robust and beyond doubt.

This brings us to the reliability of our findings regarding the 40 genes identified here. Historically, due to technical limitations, only a limited number of genes (with high mutation frequencies) have been investigated for positive selection in germline using standard sequencing. However, given the increasingly long lists of genes under positive selection in somatic tissues, we believe that it is not surprising that as we look across the exome of sperm using the highly sensitive NanoSeq method for the first time, additional genes are uncovered. But to what degree can we have confidence that the 40 genes we identified are likely to be part of the true set? Below we provide several lines of evidence supporting the robustness of our findings:

- a. Validation of method: In addition to validating the mutation rate of targeted/exome NanoSeq, Lawson et al., 2024 used blood and buccal samples to demonstrate the method's ability to identify genes under positive selection. Specifically, in blood targeted NanoSeq in 371 individuals, they identified 14 genes under significant positive selection, where all 14 are known clonal haematopoiesis drivers including the most well known genes such as *DNMT3A* and *TET2*. Importantly, the depth and sensitivity of NanoSeq revealed ~100-200 times more driver mutations compared to standard sequencing approaches. Similar to blood, in buccal samples, NanoSeq uncovered a rich driver landscape, identifying 49 genes under positive selection. Key NanoSeq driver genes in buccal, with high mutation frequencies, such as *NOTCH1*, *TP53*, *FAT1*, *NOTCH2*, and *PPM1D* align with driver mutations previously reported in tissue-relevant cancers (head and neck, skin and oesophagus). Collectively these findings demonstrate that both targeted and exome NanoSeq not only replicate findings from standard methods across different tissues but also provide exceptional resolution for studying selection, particularly in polyclonal tissues/cell types such as sperm.
- b. Independent replication: At the moment the novelty and power of the targeted NanoSeq approach means that no other technology currently exists where the scale and accuracy of sperm sequencing are achievable. For instance, the HiDEF-seq method (Liu et al. 2024) also gives the expected mutation rate in sperm, but does not exist in a targeted version and is more difficult than NanoSeq to scale due to its long-read approach. The other existing approach would perhaps be TwinStrand duplex sequencing but this method is both a) shown in two sperm sequencing studies (Kunisaki et al. 2024; Axelsson et al. 2024) to generate mutation rates too high to be consistent with trio DNM rates and NanoSeq/HiDEF-seq rates and b) no longer commercially available. While sperm sequencing on the scale of NanoSeq may not currently be possible with other experimental methods, there are key results from previous and new work

which we outline here that provide independent replication of many of the genes we infer to be under germline selection from alternative sources of germline mutations:

- i. Of the 40 genes identified as under positive selection in this study, 9 overlap with the 13 previously established germline driver genes. This firstly means that 9 of our 40 genes are already robustly validated by one or more alternative methods in the body of work establishing germline selection by the Goriely group and colleagues (Rebuttal Table 1). Secondly, given the search space of over 18k genes (albeit with extra power at many of these genes included in our targeted panel of 273 genes), it would be extremely unlikely to have such concurrence by chance in genes driving selection.
- ii. 25/40 genes have strong evidence of being validated of positive selection/driver gene properties in somatic tissues and/or cancer as evidenced by their classification as a 'Tier 1' cancer gene census gene (Sondka et al. 2018), including 16 genes that are not among the 9 established germline driver genes described above (**Rebuttal Table 1; Manuscript Fig. 2c**).
- iii. We detail an additional 4 genes in our manuscript that have not been robustly associated with disease, but have previously been shown to exhibit excess numbers of variants of the same class in the general population in our gnomAD analysis (**Rebuttal Table 1; Manuscript Fig. 4a**), as would be expected for variants under positive germline selection that are not associated with a dominant fitness cost.
- iv. We highlight a recent study: Seplyarskiy et al. 2025, which independently identified many of the same genes reported here. Shortly after the submission of our work, we were contacted by another group investigating germline driver mutations inferred from trio-based DNM whole-exome sequencing datasets. Their approach was specifically designed to separate the effects of mutation enrichment due to disease ascertainment from germline positive selection, as many trio DNM cohorts are recruited in the context of developmental disorders. Ideally, such studies would instead use DNM data from birth cohorts, eliminating the need to account for disease ascertainment; however, large datasets of this type are not yet available. Using their methodology, Seplyarskiy et al. identified 11 genes likely under positive selection that overlap with our set of 40 genes. Notably, this includes 7 genes not previously reported among the 13 genes already known (see Rebuttal Table 1). We note that two authors of this manuscript (M.N. and R.R.) are co-authors of the Seplyarskiy et al. study, contributing to analyses that supported their findings using sperm sequencing data; however, these data were not used to initially identify the significant genes.

- v. Together the lines of evidence shown above and summarised in **Rebuttal Table 1** lend additional support to our findings for 28/40 significant genes. Among the remaining 12, many have support from other angles. For instance *CUL3*, *RIT1*, and *RASA2* are all known members of the RAS-MAPK pathway, *CSNK2B*, and *CCAR2* are involved in *Wnt* signalling and *BMP2* is involved in TGF- β /BMP signalling - all pathways with genes that have definitive or highly convincing roles in germline selection. *SCAF4*, *KMT2E*, and *DHX9* are genes with links to epigenetic modification or RNA metabolism that are known to cause developmental disorders, similar to many of the genes with the strongest support in our dataset (e.g. *KDM5B*, *KMT2E*, and *DDX3X*). Even a gene with no strong links to cancer or germline selection pathways such as *PRRC2A* has been shown in mice to promote the transcriptome transition from spermatogonia to spermatocytes (Tan et al. 2023), giving a highly plausible mechanism for how loss of the gene could favour clonal expansion of spermatogonia.

Regarding the reviewers' concerns about the limitations of our study, we would like to highlight that the cohort studied is a population sample with no specific ascertainment bias e.g. for age, or disease. While additional samples from this, or another population cohort would always be valuable to increase power of a meta-analysis, we are not convinced that including another cohort would significantly enhance the strength or novelty of our findings. Moreover, we believe it is important to contextualise the resources allocated to generate this dataset compared to standard sequencing approaches. Although the sample size may appear modest, the depth of sequencing achieved in this study is substantial. For example, for exome NanoSeq alone, we sequenced 38 individuals to an average duplex coverage (dx) of ~550.61. To achieve this coverage we used 84 lanes of NovaSeq 6000. This effort is equivalent to sequencing approximately 19,320 exomes at 50x coverage using standard methods.

In summary, given: (i) the strong concordance with the established germline mutation rate and spectrum, (ii) the lack of an alternative sequencing technology with the required scale and accuracy, (iii) the strong concordance of the identified genes under germline selection with previous results in independent studies in germline and somatic tissues, (iv) the application to a population cohort without any disease or health-related ascertainment, (v) the scale of the sequencing resources deployed already and (vi) previous studies that applied the same NanoSeq technology to somatic tissues making robust findings, we do not consider that there is a strong justification to pursue replication using the NanoSeq technology in additional population-based samples, as this would not be likely to uncover substantial additional insights or add additional statistical robustness. Overall, we hope that the reviewer is given confidence in the results by the detailed validation of the NanoSeq technology and the independent replication of our key results.

Gene	Reported selfish selection	Cohort level analysis	gnomAD outliers	Tier 1 Cancer gene
BRAF				
FGFR3				
SMAD4				
PTPN11				
KRAS				
HRAS				
FGFR2				
RAF1				
CBL				
SMAD6				
PPM1D				
MIB1				
KDM5B				
CTNNB1				
NF1				
PTEN				
LZTR1				
SSX1				
AR				
ARHGAP35				
ARID1A				
KDM5C				
DDX3X				
KMT2D				
EP300				
NSD1				
TCF12				

Rebuttal Table 1 | 28 of 40 significant genes are independently replicated by alternative methods and cohorts from the work of reported selfish selection (Wood and Goriely 2022), cohort level analysis (Seplyarskiy et al. 2025), by our gnomAD analysis (Manuscript Fig. 4a), or supported as a Tier 1 COSMIC cancer gene census gene.

R1.2 Unclarity: It would be useful to describe early in the results section how mutations were called in sperm samples and how comparison to matched blood samples helped in distinguishing inherited from new mutations. In line 76 the authors mention 6,653 SNVs detected in sperm, I assume these are the SNVs not present in blood, but this is not clearly described in the manuscript.

Thank you for the suggestion, we have amended the results text as follows:

Manuscript lines 75-78: “We identified single nucleotide variants (SNVs) and small insertion–deletion mutations (indels) in whole genome NanoSeq data from sperm and blood. Inherited germline variants were excluded using matched standard whole-genome sequencing data from blood, and stringent variant filtering was applied (Methods).”

R1.3 Overall, the study design is complex and involves the analysis of different sample numbers for different approaches, including three different targeted gene panels as well as exome sequencing. It is very difficult to understand what was done, what the concordance was between the different approaches and understand the reliability of the approaches. This is further complicated by the identification of DNA contamination which was acknowledged by the authors. While the authors claim they were able to in silico decontaminate their results, it does make one wonder about the overall robustness of the data (hence the request for independent replication, see above).

We appreciate this feedback regarding complexity and have created a summary graphic to better explain the differences and concordances between approaches and sample counts now added as **Extended Data Fig. 2a** of the manuscript also shown below as **Rebuttal Fig. 2a**.

All sequencing data presented falls under the NanoSeq method, which should all have accurate and concordant mutation rates. We detail in response to the reviewers first comment the validation of the mutation rate accuracy, while **Rebuttal Fig. 2c** demonstrates that after correcting for trinucleotide composition and driver genes, all sequencing methods generate highly concordant mutation rates.

The largest difference in approaches is that between the genome and both target panel approaches (“targeted” and “exome”). These differences are summarized in **Rebuttal Fig. 2a** and include the method of DNA fragmentation, the coverage breadth and depth (shown in detail in **Rebuttal Fig. 2b**) and the analyses for which they were applied. In particular the genome approaches are only used for mutation burden and mutational signature analyses, confined in the main text to **Fig. 1** and the first 4 paragraphs. The remainder of the main text and main figures are restricted to the target panel approaches which were used for selection, driver, and pathogenic variant analyses due to their high depth and concentration in coding regions.

The target panel approaches are effectively identical in reliability and concordance with the only difference being which regions were used in a target capture panel to pull down DNA in library preparation. The reviewer mentions “three different targeted gene panels as well as exome sequencing”. We would clarify two points here: firstly the exome NanoSeq approach is one of the target panels rather than being standard exome sequencing. Secondly, while there were technically 3 total panels (“targeted”, “exome”, and “targeted pilot”), the “targeted” and “targeted pilot” were so similar that they were merged in the data processing stage, as described in the **Targeted gene panels** section of the methods, meaning that they were able to be treated as the same panel.

We have attempted to provide further clarification of sample counts used with the different methods in the summary graphic in addition to counts available in **Supplementary Table 1**. These counts were influenced by many factors such as DNA yield during extractions, sequencing budget, sequencing success, and passing QC.

Overall we hope that the addition of **Rebuttal Fig. 2a** along with **Rebuttal Fig. 2b,c** and summarizing data in **Supplementary Table 1** make the differences and concordances between the 4 sequencing sets in the paper (“Blood genome”, “Sperm genome”, “Sperm targeted”, “Sperm exome”) clearer for the reviewer and future readers.

Rebuttal Fig. 2 | Method and coverage summary a, Graphical relationship between NanoSeq methods applied in manuscript. Both sperm and blood samples underwent genome NanoSeq, used for mutation burden and mutational signature analyses. Only sperm samples were used for targeted NanoSeq applied for selection, driver, and pathogenic variant analyses. Targeted NanoSeq is adaptable to different target panels, and we have named the sample sets as “targeted” for the samples using the 263 gene cancer panel and “exome” for samples using the exome wide panel. **b**, Mean duplex coverage (log scale) and percentage of genome covered (log scale) per sample. Panels summarise the mean duplex coverage (dx) and mean percentage of genome covered per NanoSeq type and tissue. **c**, Mutation burden of targeted (dark orange), exome (yellow), and genome (blue) sperm sequenced samples that are observed without correction (left), corrected for trinucleotide composition of covered base pairs relative to the whole genome (middle) or corrected and masked for mutations and coverage in the 44 genes linked to germline positive selection (right). Model fits are linear regressions with 95% CI bands.

Regarding the reviewer’s concerns related to *in silico* DNA decontamination, as outlined in the methods section, the *in silico* decontamination was applied due to the 12 samples (3 exome panel and 9 targeted panel) that exceeded our predefined contamination thresholds. We mention in the Methods section that the decontamination process effectively alleviated the identified small excess mutation burden and the high ratio of masked SNPs to passed variants. This observation suggests that the contamination issue was successfully mitigated, preventing these samples from skewing the overall results. To further confirm that contamination did not compromise our findings, we conducted an additional set of analyses excluding these 12

samples entirely. Below, we summarise the key metrics comparing the original analysis with the post exclusion analysis:

1. Sample/variant count
 - Manuscript: 38 exome (56,503 variants), 81 targeted (5,059 variants)
 - Exclusion: 35 exome (51,898 variants), 72 targeted (4,538 variants)
2. Mutation burden (**Rebuttal Fig. 3a**)
 - Manuscript: exome: $7.5 \times 10^{-10} \times age + 5.9 \times 10^{-10}$, targeted: $7.7 \times 10^{-10} \times age - 2.5 \times 10^{-9}$
 - Exclusion: exome: $7.3 \times 10^{-10} \times age - 2.8 \times 10^{-11}$, targeted: $8.9 \times 10^{-10} \times age - 4.1 \times 10^{-9}$
3. Exome-wide dN/dS ratio (**Rebuttal Fig. 3b**)
 - Manuscript: 1.07 (95% CI: 1.04-1.10)
 - Post-exclusion: 1.07 (95% CI: 1.03-1.10)
4. Significant gene list
 - Manuscript: 40 genes
 - Post-exclusion: 39 genes (loss of *CTNNB1*)
 - i. *CBL* shifted from exome-wide significant to restricted hypothesis significant
 - ii. *CTNNB1* shifted from restricted hypothesis significance ($q=0.09$) to not significant ($q = 0.13$).
 - iii. Of the 25 significant hotspots, 16 remained post-exclusion, but all lost hotspots were within genes with either another significant hotspots or reaching gene-level significance. No further genes were removed from the overall significant list.

Overall, these results confirm that the exclusion of the 12 samples leads to only minor changes in our findings, primarily reflecting a small loss of statistical power rather than any systematic bias or skewing introduced by contamination. Key results including mutation burden, exome-wide dN/dS ratios, and the significant gene list remain consistent with those presented in the manuscript.

Rebuttal Fig. 3 | Excluding decontaminated samples a, Mutation burden of exome (left) and targeted (right) sequenced samples with the 12 decontaminated samples marked in yellow. Linear model fits are shown with a dashed black line for the fit presented in the manuscript (Extended Data Fig. 2b - Observed) and as a dark blue line for the fits after excluding the 12 samples. **b**, Comparison of global dN/dS values from the 38 exome sequenced samples presented in the manuscript (blue) or with only the 35 exome sequenced samples post exclusion of decontaminated samples (red). Categories are all nonsynonymous mutations, missense, nonsense or essential splice. Error bars indicate 95% CIs.

R1.4 Details: Unclearities very much relate to different sample numbers used and referred to throughout the manuscript. After reading the manuscript a few times I am still not clear on some numbers referred to in the manuscript. The results section (line 64) refers to 81 semen samples for which whole genome Nanoseq was done, whereas supplementary note 1 it seems like sperm counting was done in 85 samples of which 6 were excluded, so 79 in total? Also, line 67 talks about that these sperm samples and matched blood comes from 63 men, whereas Suppl. Table 1 mentions 62 men, and only 57 passed analysis.

Thank you for noting these inconsistencies. In the instance related to sperm counting all these numbers are correct, but there is a detail missing which we have added to **Supplemental Note 1**.

Supplement lines 75-77: “There were two samples for which we were unable to obtain sperm counts due to exhaustion of the sample during DNA extraction, but which were retained due to having non-outlier mutation burdens and sperm counts from another timepoint of that individual within normal count ranges.”

We have also verified that the inclusion of these samples doesn’t substantially impact the mutation burden estimate (new text in italics):

Supplement lines 79-87: “The mutation burden estimate in sperm presented in Figure 1 of the main text was 1.67 haploid single base substitution (SBS) per year (95% confidence interval (CI) = 1.41-1.92, linear mixed-effect regression) with an intercept of 3.0 haploid SBS (95% CI = -10.3 to 16.4). This estimate is not substantially impacted by excluding [...] *or by excluding the two samples without sperm counts: 1.68 haploid SBS per year (95% CI = 1.42-1.94, Poisson regression) and intercept of 2.5 haploid SBS (95% CI -11.03-16.05).*”

In the second instance, our original sentence does imply that the analyzed sperm and blood samples each had 63 independent donors, but this is not the case for sperm. We have amended the sentence to correct this: (added text for per tissue donor count).

Manuscript lines 64-67: “We performed restriction enzyme based, whole genome NanoSeq²³ of bulk semen samples (n = 81; 1-2 timepoints per donor; 57 donors; age range: 24-75 years) and matched blood (n = 119; 1-3 timepoints; 63 donors; age range: 22-83 years) from men in the TwinsUK cohort²⁶ (including 8 monozygotic and 3 dizygotic twin pairs; Methods; Supplementary Table 1).”

Referee #2

This manuscript by Neville et al detected de novo mutations in 81 sperm samples from 63 men in the TwinsUK cohort by using a high-accuracy single-molecular duplex sequencing technique (NanoSeq). Based on excess non-synonymous mutations relative to synonymous mutations, the authors detected evidence of positive selection across the exome as well as on ~40 genes. As many of these genes are known to be associated with cancer predisposition and developmental disorders, the authors estimated that disease-causing mutations are enriched in sperm by 2-3 fold, driven by the competitive advantages that they confer during spermatogenesis.

Overall, the paper is well written and showcases the combined strengths of several groups in various fields of mutation study (e.g., cancer genomics, germline mutagenesis, mutational basis of developmental disorders, and technology and computational method development). As the baseline (neutral) mutation model is central to the selection test, my only major concern is the sensitivity of the results to the mutation model, and I also have some questions regarding the mutation model itself. Additionally, I noticed that the authors planned to make the raw sequencing data available on EGA (with accession number pending) but not the sperm mutation calls. Since these sperm mutations are technically not part of the individual genotypes, could the authors release these mutation calls in the supplementary tables?

Unless this data release is prohibited by consent or other regulations, I think these data should be released in accordance with Nature's policy.

We thank the reviewer for their helpful comments and constructive feedback.

Regarding release of mutation calls, we have asked the TwinsUK committee, and they have now agreed to release these mutations upon publication: The committee "...have approved releasing to the journal, if required, the name and number of the rare mutations that have occurred within the cohort, but without links to individuals, IDs or identifiable information. Access to which variants occurred in the same individual cannot be freely released, but must be applied for through managed access (effectively this is similar to how we release SNP frequencies in the cohort, but individual genotype data is under managed access)."

The anonymised variants are now available as **Supplementary Tables 2,3, and 7**. We note that managed access to the full data including IDs, ages, and detailed variant annotations is already available, and one collaborator has successfully accessed the data via TwinsUK.

Major comments:

R2.1 Inference of selection

R2.1.a As Martincorena et al (2017) demonstrated, the use of imprecise mutation models can lead to systematic bias in estimates of dN/dS and incorrect inference of selection. In this manuscript, the authors went to great lengths to improve the mutation model, including modeling base-specific sequencing coverage, methylation level and 5-mer sequence context. Extended Figure 6d showed that each of these refinements of the model had non-negligible impact on the estimate of dN/dS ratios. This raises a natural question: is the current model sufficiently precise that further refinements (such as 7-mer context or finer groupings of methylation levels) would not lead to further downward adjustment in the dN/dS estimates? Is there empirical data (i.e., adding additional features do not change the results) or semi-quantitative argument to address this possibility?

This is a good point, in the literature there certainly exists evidence of mutation rate heterogeneity beyond the 3 methylation level bins or 5mer context or the chosen here (Aggarwala and Voight 2016; Carlson et al. 2018). We detail our justifications for stopping our exploration of models beyond these boundaries below. One small point we'd mention is that any difference in dN/dS values from model changes would not necessarily be downwards. As seen for splice and nonsense variants in **Extended Figure 4d** (included below for reference), these adjustments can be upwards, it just so happens that each refinement we investigated brought the overall dN/dS estimate downwards, largely because the biggest class of variants, missense variants, were revised downwards in each adjustment.

7-mer context: There is convincing evidence that there exists heterogeneity beyond 5-mer context in germline mutations (Aggarwala and Voight 2016; Carlson et al. 2018). While it would be reasonable to explore the effect of 7-mer context, our dataset of 32,386 coding SNVs lacks the statistical power to support such a model.

For each jump in context (eg., 3mer to 5mer) the number of mutation contexts increases 16-fold and thus the average number of mutations per context and power to estimate mutations rates drops 16-fold:

3-mer + 3 CpG meth bins: $3 \times 4^3 + 2 \times 2 \times 4^1 = 208$ contexts (156 mut per context)

5-mer + 3 CpG meth bins: $3 \times 4^5 + 2 \times 2 \times 4^3 = 3,328$ contexts (9.7 mut per context)

7-mer + 3 CpG meth bins: $3 \times 4^7 + 2 \times 2 \times 4^5 = 53,248$ contexts (0.61 mut per context)

Equation explanation: calculating total contexts with CpG meth bins is given by: $3 \times 4^X + (Y - 1) \times 2 \times 4^{X-2}$

- 3 is from the 3 possible mutations given a starting base pair
- X = mer value (number of base pairs where any of the 4 nucleotides are possible)
- Y is total methylation bins, subtract 1 because one of that CpG context is already in the standard contexts
- Instead of having X free base pairs, the reference base can only be C or G so using one 4 multiplier becomes a 2 and then given a C or G the following bp must be a G or (reverse complement) the preceding must be C, meaning that bp is fixed and a second 4 multiplier becomes a 1.

This means that were we to use a 7mer model, we'd have fewer training variants than contexts and many 7mer contexts would not have an observed variant. In contrast, the previous publications which investigated 7mer contexts had millions of training variants (e.g. Carlson et al. 2018 used 36 million singleton polymorphisms for training their model).

To illustrate the reduced power of more complex models, we compared the detection of significant changes in mutation rate between 3- and 5-mer contexts. For the 3-mer model, there were 208 possible contexts, of which 33 had significant heterogeneity when extended to the 5-mer level (**Supplementary Note 3; Supplementary Table 9**). The smallest number of mutations in any 3-mer contexts where we detected significant heterogeneity at the 5-mer level was 87 mutations. If we take this as a rough estimate of our power limit, we could say that although we detected 33 total 3-mer contexts with significant heterogeneity, 52% (109/208) had enough mutations (>87 mutations per context) to detect a significant effect. However, considering the next step up: 5-mers to 7-mers, only 1.5% of the 3,328 5-mer contexts meet the threshold of >87 mutations, suggesting that we would only be powered to investigate the possibility of 7-mer heterogeneity in a small proportion of contexts.

Overall, while we believe using 7mer contexts is worth considering in the future for larger datasets, we do not believe we are powered to interrogate them here. We added the following text to **Supplemental Note 3** to reflect this.

Supplement lines 236-238: "The size of the dataset (32,386 coding SNVs) did not allow for interrogation of heptanucleotide context, for which the opportunity matrix would increase to $53,248 \times 4$, an average of only 0.61 mutations per context."

Methylation bins: In contrast, for methylation, extending from 3 bins to, for example, 10 bins creates about 900 new contexts. This is not a negligible amount but few enough that we would plausibly be powered to investigate to this level, especially since CpG sites often have a large proportion of mutations.

5-mer + 3 CpG meth bins: $3 \times 4^5 + 2 \times 2 \times 4^3 = 3,328$ contexts (9.7 mut per context)

5-mer + 10 CpG meth bins: $3 \times 4^5 + 9 \times 2 \times 4^3 = 4,224$ contexts (7.7 mut per context)

In this case, the reason we did not implement a model with more than 3 bins comes back to why we investigated possible mutation rate biases to control for in the first place: any variable which impacts mutation rate and is not evenly distributed between synonymous and non-synonymous variants can bias dN/dS values. Methylation level met these conditions: CpG non-synonymous variant classes were on average 8.1% more methylated than synonymous variants (**Extended Fig. 4b; Rebuttal Fig. 4a**) and methylation level impacted mutation rate (**Extended Fig. 4c**). We found that after adjusting using the 3 methylation bins this bias is largely resolved, with an average absolute difference in methylation level % between synonymous and non-synonymous variants dropping from 8.1% to 0.24% (**Rebuttal Fig. 4a**). Extending methylation to the example of 10 bins split by decile would reduce this bias even further, but only by a very small amount: 0.24% to 0.07% (**Rebuttal Fig. 4b**). We therefore reason that the additional ~900 contexts created by extending methylation bins, though perhaps feasible, was not likely to add meaningful adjustments to the dN/dS values given that just 3 bins seems to largely resolve the bias. We added the following text to **Supplemental Note 3** to explain this:

Supplement lines 219-225: “This correction reduced the average absolute bias between synonymous and non-synonymous CpG sites from 8.1% to 0.24% and slightly lowered the observed dN/dS ratios as expected (Extended Data Fig. 4d). We found that using finer methylation bins, such as 10 bins with one bin per methylation percentage decile, provided only a marginal additional reduction in bias between synonymous and non-synonymous CpG contexts: from 0.24% to 0.07%. We therefore used only the 3 methylation bin correction as we reasoned that the potential power loss from introducing additional contexts outweighed this minor improvement.”

Extended Data Fig. 4 | Model selection dN/dS **a,b**, Mean duplex coverage (**a**) and methylation percentage (**b**) of all base pairs with exome sequencing coverage split by mutation consequence. **c**, C>T mutation rate at CpG sites in exome sequenced samples split by methylation bin based on percentage methylated from testis bisulfite sequencing⁷⁰. **d**, Comparison of global dN/dS values from exome sequenced samples using different modifications to the *dNdScv* algorithm. Categories are all nonsynonymous mutations, missense, nonsense or essential splice. The basic model excludes genes which have no coverage but otherwise uses default parameters. Additional models show the impact of adding corrections for duplex coverage per base pair (BasePairCov), CpG methylation level (CpGmeth), and pentanucleotide context (Penta). **e**, Comparison of per-gene significance in exome-wide (blue) or restricted hypothesis (orange) dN/dS tests using the different models. Genes that did not reach significance in either test are shown in grey. Error bars indicate 95% CIs.

Rebuttal Fig. 4 | Methylation bias for different methylation level bins a,b, The percentage difference in methylation of non-synonymous variant classes relative to synonymous variants under a 3-bin (a) or 10-bin (b) split with the uncorrected difference (“None”) for context.

R2.1.b It is surprising that a significant signal of selection could be detected, given that the vast majority of mutations (99.5%) were seen in only a single duplex molecule. Such mutations in the sperm pool are analogous to singletons in polymorphism datasets. Extremely rare variants are expected to deviate little from the mutation input as there is limited opportunity for selection to influence their trajectories, and I suspect that the dN/dS ratio for extremely rare polymorphisms would not deviate significantly from neutral expectation, despite the large amount of data. Is the additional power due to the intermediate coverage per site (which is analogous to sample size in polymorphism dataset)? Or is it due to the special clonal structure of seminiferous tubules? Can the authors perform power simulations to demonstrate that it is feasible to detect signal of selection in rare variants?

We appreciate the reviewer's question and the opportunity to clarify why it is not surprising that we are powered to detect selection in sperm, despite most mutations being observed in only a single duplex molecule. While there may be some similarities between mutations under selection in sperm with singletons in population datasets, we believe this analogy may not hold true for this specific point due to fundamental biological differences in the selection pressures and contexts governing mutations in these settings.

Selection pressures pre- vs. post-conception: the key distinction between sperm mutations and population-level singletons is the selection environment. In the germline context, the dynamics of selection occurs entirely within testes, where clonal expansion of spermatogonial stem cells (SSCs) allows the selection to act at the cellular level on hierarchy of stem cells, progenitors and committed cells, prior to fertilisation. This is markedly very different from population level singletons, where mutations are filtered through multiple steps: germline transmission, embryogenesis, development, and survival to beyond reproductive age. In population datasets, selection acts overwhelmingly post-conception, with negative selection dominating due to constraints on organismal fitness (as a side note this negative selection is in fact detectable at the level of singletons, see for instance **Fig. 4a** in this manuscript). In contrast, in reproductive tissues positive selection often dominates, favouring mutations that enhance SSC proliferation during ageing, very similar to other somatic tissues. This phenomenon has been consistently observed in other proliferating tissues: studies of hematopoietic stem cells have detected evidence of selection acting on rare variants, including those observed at low clonal fraction (Jaiswal et al. 2014; Martincorena et al. 2015). Similarly, evidence of positive selection has been found in clonal expansions of rare mutations in oral epithelial tissues (Lawson et al., 2024).

Histological structure of testes: the reviewer raises an interesting point about whether the unique histological structure of the seminiferous tubules, such as their tubular architecture, semi-open stem cell niche, partially one-dimensional structure, and separation from other tubules, might influence clonal expansion and clonal size in testes. This hypothesis supports the highly polyclonal nature of the testes and suggests that these structural features could regulate selection dynamics, potentially limiting the physical expansion of clones for genes under selection. There is some interesting work in mice that hints at this possibility (Yoshida 2020). However, future work in this space is certainly needed to better understand how these physiological pressures shape clonal dynamics in this tissue. This remains an exciting area for future exploration.

R2.2 Details of dNdScv algorithm and its application to data

Although Martincorena et al (2017) clearly explained and benchmarked the dNdCcv algorithm, it would still be helpful if the authors could briefly outline the model for inferring the background mutation rates, annotating variant effect, and clarify exactly how the model is applied to the data.

To address the reviewers request we have added the following sections to **Supplemental Note 3** and created **Supplementary Table 10**, which summarises how the model is applied to each dN/dS analysis in the manuscript.

Supplement lines 131-139: “Overview: Positive selection in sperm was quantified by estimating the rate of non-synonymous (N) relative to selectively neutral synonymous (S) mutations (dN/dS ratio). All dN/dS results were derived using the dNdScv package⁴. The general framework of these analyses is that 1) a set of mutations for analysis are selected; 2) these mutations are annotated for their gene/consequence; 3) a mutation model is learned from these input mutations; 4) the dN/dS value for a set of genes/single gene/single hotspot is calculated. The specifics of how these steps were conducted for each dN/dS analysis are described in detail below, summarized in **Supplementary Table 10**, and have the relevant code available at <https://github.com/mattnev17/spermPositiveSelectionManuscript>.”

Supplement lines 149-155: “Variant Annotation: Mutations input to *dndscv* were annotated with the method’s default GRCh37 reference transcript set. This transcript set was originally generated by choosing the longest complete coding sequence per gene from BioMart as described on github: (<https://htmlpreview.github.io/?http://github.com/im3sanger/dndscv/blob/master/vignettes/buildref.html>). Using the transcript set, mutations are then annotated as synonymous, missense, nonsense, essential splice, or coding indels based on an internal codon/essential splice model⁴. No coding variants in the 40 significant selection genes have an alternative interpretation in *dndscv* when using the canonical (rather than longest) coding transcript.”

Supplement lines 294-304: “Age bracket dN/dS: Calculation of exome-wide dN/dS ratios for each of the three age brackets (Fig. 2f) used the same dNdScv method applied to the full dataset calculations but fit a new mutation model for each age bracket to account for possible differences in mutation types over age. Specifically, exome sequenced samples were divided into three equal age groups of 26-42 year olds, 43-58 year olds, and 59-74 year olds and each mutation set was input into the custom dNdScv basePairCov + CpG meth + Penta model described above.”

R2.2.a It is unclear how the effect of a variant in a gene is determined. Specifically, a variant may have different effects on different splicing isoforms. How are these differences reconciled and integrated into a single effect?

As described in Variant Annotation above, *dndscv* uses the longest coding sequence transcript per gene. This approach is a limitation of *dndscv* because any generic rule for selecting a transcript (in this case, the longest coding sequence transcript) will not always be the most appropriate one for the tissue under investigation. We can confirm that in our particular case, no coding variants in the 40 significant selection genes have an alternative interpretation in *dndscv* when using the canonical transcript denoted by Ensembl.

R2.2.b The application of the dNdScv algorithm was insufficiently described. For example, for results shown in Fig 2f, it is unclear whether three mutation models were inferred separately based on mutations observed in each age group, or if only one mutation model was inferred based on all sperm mutations identified, with the same model being applied to calculate dN/dS ratios of observed mutations in the three age groups. It has been shown that the mutation spectrum of germline mutations significantly varies significantly with age at reproduction for both parents (for example, CpG>TpG mutations are more enriched in DNMs passed down from older fathers), which suggests it may be necessary to infer the background mutation model for each age group. Related to this, do the authors observe correlation between the fraction of SBS1 mutations with age?

We agree with the reviewer and for the reasons stated, we did infer a new mutation model for each of the 3 age groups when calculating dN/dS ratios. This is now properly described in Age bracket dN/dS from **Supplemental Note 3** and **Supplementary Table 10**.

In regards to SBS1 rate, to our knowledge, there are conflicting results as to whether mutation types such as CpG>T mutations are enriched relative to other types in older fathers or not. For instance, one recent trio publication showed an enrichment (Fig. 5 in Shojaeisaadi et al. 2024), while another showed a depletion (Fig. 2S2 in Sasani et al. 2019). Our results presented in **Extended Fig. 2b** of the manuscript finds an increase with age of both SBS5 and

SBS1. Converting this to fractions vs age we find no significant relationship between the ratio of SBS5 to SBS1 vs age ($p = 0.48$; **Rebuttal Fig. 5**).

Rebuttal Fig. 5 | Signature fraction vs age The ratio of mutation signatures SBS5 to SBS1 vs age where each point is a sperm sample sequenced with genome NanoSeq. Model is a linear mixed regression with 95% CIs calculated by parametric bootstrapping.

R2.2.c Similarly, it is unclear whether gene expression level was explicitly modeled as a feature and whether the same or different mutation models were used for estimating dN/dS for mutations in genes of different expression levels. If only one mutation model based on all mutations was inferred, the high dN/dS ratio of highly expressed genes (Fig 2g) may reflect deviation in the mutation profile of these genes from the exome average due to transcription-associated molecular processes.

This is a good point that we did not originally consider. As described in the *genesetdnds* function of the *dndscv* package there are two possible approaches for calculating the dN/dS values of a subset of genes

- A. Training the mutation model on the entire dataset then testing within the gene set of interest. This has the advantage of more power for training the mutation model.
- B. Training the mutation model and testing within the gene set of interest. This has the advantage of training the model within the feature if it is suspected that mutation processes might differ from the wider dataset.

In the manuscript we took the first approach (A), which is now explicitly stated under Gene set enrichment in **Supplementary Note 3** and in **Supplementary Table 10**:

Supplement lines 309-311: “This means that these gene sets used the mutation model inferred for the whole dataset, then applied it to the gene set of interest, rather than learning a new mutation model for each input.”

The reviewer is correct however that it is possible that transcription-associated molecular processes might be different for different expression levels, rendering approach A inaccurate and approach B more appropriate. We investigated this possibility but found that the results from both approaches were largely consistent (**Rebuttal Fig. 6**). As expected, approach B (blue points) had wider confidence intervals expected of less training data. Given the similarity of results we have elected to remain the approach taken in the manuscript for consistency with other gene set tests.

Rebuttal Fig. 6 | Expression gene set dN/dS model dN/dS ratios for sperm SNVs across expression sets of genes, where the dotted black line indicates neutrality and the dotted orange line represents the cohort average across all genes. Expression levels represent 7 bins of mean expression levels across germ cell stages and expression clusters represent genes most characteristic to certain germ cell stages. Colour indicates the training set of mutations for the underlying mutation model where exomeWideInput (red) indicates that the mutation model learned rates from all exome-wide mutations whereas geneSetInput (Blue) indicates that the mutation model learned rates from only on mutations in the specific geneset.

Minor points:

R2.3 Line 24 (abstract): the sentence “while four genes that exhibit...” is grammatically problematic.

Removed “that” (Manuscript Line 24).

R2.4 Line 33: “replicating (tissues)” should be “dividing” or “proliferating”.

Replaced “replicating” with “proliferating” (Manuscript Line 33).

R2.5 Line 66: the authors mentioned 9 pairs of monozygotic twins in the main text, but 8 pairs in the supplemental note 2. Why the difference?

Thank you for flagging this. The number 8 is correct for both and we've changed the main text from 9 to 8 (Manuscript Line 66). We accidentally put the pre-QC number in the main text (**Supplementary Table 1**).

R2.6 Lines 323-324: The phrase "on known driver genes" should be added after "germline positive selection". Otherwise, it seems to imply other non-selection mechanism could drive the higher than expected rate of disease-causing mutations.

We have added the suggested text "on known driver genes" (Manuscript Line 329).

R2.7 Line 691: Should the word "multiplying" be "dividing"? Alternatively, the ratio should be "the total genome size to the sample's callable genome".

This is correct, we have changed the order of the described ratio as suggested and added in the equation for clarity. We can also confirm that the code for the calculation was done in the correct way and not in the way described in the methods.

Manuscript lines 736-738: "...multiplying the paternally phased DNM count by the ratio of the total genome size to the sample's callable genome size: $DNM_{\text{paternal}} \times (\text{total_genome}/\text{callable_genome})$."

Referee #3

The manuscript presents a compelling and technically well-executed study that provides significant implications into reproductive biology and the dynamics of germline mutations, particularly in the context of paternal age and disease genetic risks. Utilising the duplex sequencing method NanoSeq, the authors identify positively selected genes in spermatogonia by sequencing bulk sperm (and blood as a "control"), identifying 40 genes under positive selection in the male germline linked to developmental disorders or cancer predisposition disorders in children. Their analyses indicate that the dominant selection force on germline mutations is positive selection during spermatogenesis but negative selection between generations. The findings presented, and the far reaching implications of 3-5% of sperm from older individuals harboring pathogenic mutations, represent a very significant advance and this is a well-written and timely study. NanoSeq appears to be the right method for the study presented; a relevant update to NanoSeq (capture-based) is employed and the authors convincingly show how this enhances the power for identifying mutations positively selected in spermatogonia. Still there are several areas where the manuscript could be strengthened to enhance clarity, depth, and rigor.

R3.1 Duplex sequencing coverage seems adequate but somewhat on the low side. It would be helpful to detail coverage statistics across repeat classes and discuss limitations,

particularly in segmentally duplicated regions and STRs. The authors should discuss, and quantify, the likely under-representation of reads in repeat-rich regions affecting variant calling. I wonder, would the authors even be powered to uncover evidence of (recurrent) de novo STRs? I would not necessarily expect the authors to solve this issue, but it should be spelled out clearly in the discussion.

We thank the reviewer for their insightful comments and suggestions. The mean duplex coverage (dx) of 3.67 per sample of our whole genome NanoSeq data and even the depth of our whole-exome (550.61 dx) and targeted panel NanoSeq (985.32 dx) may appear modest relative to some standard sequencing or duplex sequencing approaches. However the size of the captured regions mean that the number of base pairs surveyed and mutations called make this size of this dataset rival the largest cohorts of de novo coding mutations generated to date. For instance, as detailed in response to Reviewer 1's first question, the sequencing devoted to the 38 exome NanoSeq samples is equivalent to approximately 19,320 exomes at 50x coverage using standard methods.

While we fully agree with the reviewer that studying STRs and other mutation types is of great importance, we believe this falls outside the scope of the current manuscript due to the following technical limitations:

1- Our targeted panels (both targeted and exome NanoSeq approaches) purposefully exclude repeat-rich regions, including some STRs and segmentally duplicated regions and the analysis pipeline. Additionally, the NanoSeq pipeline requires accurate unique mapping with short-reads, meaning many repetitive sequencing regions would not progress to variant calling. We verified coverage of STR regions within our targeted panel and found that there is some coverage of these regions. For instance, using the HipSTR resource (Lundström et al. 2023) we find there is 855kb of STR associated regions with some coverage in our targeted/exome datasets, although the mean coverage in these regions is lower than the dataset average (4,934dx vs. 9,618dx).

2- The current NanoSeq analytical pipeline is optimised for SNVs and indels, and it does not support the reliable analysis of STR variants. To address this, we need to test compatibility of NanoSeq data with tools such as ExpansionHunter or STRetch which can be interesting work for future follow ups.

Overall, to address the reviewer's suggestion, we have added a discussion paragraph to provide a clear explanation of this limitation and the potential to investigate these variant classes in the future:

Manuscript line 501-506: "These results should be interpreted with the understanding that they focus solely on SNVs and small indels in sperm, without addressing other mutation or recombination events such as crossovers, structural variants, repeat-associated variants, and copy number variants. While some of these have begun to be explored in sperm using long-read sequencing (Schweiger et al. 2024), further technological advancements will be needed to assay the complete catalogue of variant classes now possible for parent-offspring trio analyses (Porubsky et al. 2024) and their contribution to germline selection and disease."

R3.2 The manuscript should address the potential biases introduced by the choice of restriction enzyme in the NanoSeq protocol. A discussion of whether different enzymes might yield varying results and how this could affect conclusions is warranted.

This is an important point regarding potential biases introduced by choice of restriction enzyme in the whole-genome NanoSeq protocol (the targeted/exome NanoSeq protocol uses sonication and not restriction enzymes). To control for this we adjust the two analyses performed with whole-genome NanoSeq (mutation burden and mutational signatures) by their trinucleotide composition relative to the full genome as specified in the methods, where we have also added text to expand on this point (new text in italics):

Manuscript lines 713-722: “Corrected mutation burdens: Given that mutation rates are strongly influenced by trinucleotide composition, it is important to consider differences in sequence composition when comparing mutation rates in datasets that target different regions of the genome. For instance, it is known that coding regions such as those in NanoSeq target panels, are biased towards a higher mutation rate partially due to a higher GC density than non-coding regions⁸⁰ which make up the majority of sequenced regions in whole genome NanoSeq datasets. *Additionally, the restriction enzyme used in whole genome NanoSeq will systematically bias sequencing coverage to specific genomic regions which may not reflect the full genome.* To correct for these effects, in each sperm NanoSeq dataset we generated a corrected mutation burden relative to the full genome trinucleotide frequencies as described previously²³.”

Manuscript lines 761-762: “Mutational signature analysis: [...] *The number of mutations were normalised for the tri-nucleotide context abundance specific for each sample relative to the full genome.*”

Reassuringly, we do not observe any significant differences in mutation rate estimates after this trinucleotide adjustment and accounting for selection effects (**Extended Data Fig. 2**). Additionally, we refer to a related preprint (Lawson et al 2024; **Rebuttal Fig. 1b**), in which the performance of restriction enzyme (RE) NanoSeq approach was compared with two sequence agnostic alternatives for DNA fragmentation: sonication method and exonuclease digestion. They have demonstrated that both the RE protocol and the alternative sonication approaches yielded highly similar mutation loads and mutational signatures, consistent with standard sequencing approaches such as single cell derived colonies. Overall, these findings support that the use of RE-NanoSeq does not introduce biases that would affect our results provided that we adjust for the base composition of sequenced regions.

R3.3 The capture based approach to yield higher duplex coverage is a well chosen experimental setup, allowing the authors to sensitively detect positive selection and revealing an increase in the dN/dS ratio with age in genes expressed during spermatogenesis. The manuscript would need to provide a detailed description of this deep whole-exome/targeted NanoSeq method

These methods are unchanged from those detailed in a companion paper describing the technique now available on medRxiv which may not have been accessible to the reviewer at the time of review (Lawson et al, 2024)

(<https://www.medrxiv.org/content/10.1101/2024.10.30.24316422v2>). To provide ease of access to these methods while respecting Nature's guidelines on methods reporting that "Detailed descriptions of methods already published should be avoided; a reference number can be provided to save space" we added the following sentence which will point any reader more directly to the relevant material:

Manuscript lines 658-659: "These steps are described in detail in *Sonication NanoSeq, Library amplification and sequencing*, and *Hybridization Capture* of Supplementary Note 1 in Lawson et al."

R3.4 The reported increase in the dN/dS ratio with age is an interesting finding (Line 153). However, it is unclear if this trend is statistically significant. Could the authors provide additional details on the statistical tests performed and include a clearer presentation of significance in Figure 2f?

Performing a linear regression, we find that the increase is not statistically significant. We have added this finding to the main text and added qualifiers to our interpretation to reflect this (new text in italics):

Manuscript lines 153-159: "Splitting the cohort into thirds by age, we find that the exome-wide dN/dS ratio was 1.01 (95% CI 0.93-1.09) in 26-42 year olds, 1.03 (95% CI 0.97-1.10) in 43-58 year olds, and 1.09 (95% CI 1.06-1.13) in 59-74 year olds (Fig. 2f). *A linear regression of dN/dS ratio against age group showed a positive trend, though statistical significance was not reached ($P = 0.18$).* This suggests that the dN/dS ratio *may* increase over male lifespan and that *if so*, the cohort-wide dN/dS ratios presented here in part reflect the age distribution of samples (age range 26-74; mean 53 years)."

R3.5 Including a gene not expressed in spermatogonial cells as a negative control in Figure 2abc would strengthen the evidence for selective pressure in genes expressed during spermatogenesis.

We present below in **Rebuttal Fig. 8** an example of 10 randomly selected unexpressed genes as well as the mean values of all unexpressed genes as "mean_unExp". In our opinion it is difficult to select a specific (or set of specific) unexpressed gene as a negative control in this figure as there is great variability in these genes - some have no mutations while others do show a (non-significant) enrichment. An alternative that remains in the spirit of the reviewer's suggestion is to include the mean values of all unexpressed genes shown by "mean_unExp". While possible we believe that the main point this would convey is that non-expressed genes are near the expected value of dN/dS = 1. This neutral expectation is already indicated in **Figure 2 panel b** with the dotted line and we think this result is perhaps more appropriately placed in **Figure 2g** alongside other results on expression levels.

Rebuttal Fig. 8 | Gene dN/dS unexpressed gene control a,b,c Genes with significant dN/dS ratios from exome-wide and restricted hypothesis tests and 10 non-significant unexpressed genes, as well as the mean of all unexpressed genes. a, Mutation count split by mutation class. b, Enrichment over expectation of mutation classes. c, Mutation type driving dN/dS enrichment, COSMIC cancer gene tier, developmental disorder gene link in DDG2P, and potential germline selection geneset.

R3.6 While age is a dominant factor influencing mutational burden, the potential effects of BMI, smoking, and alcohol consumption on mutational signatures and dN/dS ratios should be explored in greater detail. Were there subtle trends, even if not statistically significant, that might warrant further investigation in larger cohorts?

To investigate further and explore potential trends in our dataset we have explored effect sizes and their confidence intervals from our model rather than looking exclusively at significance values (**Rebuttal Fig. 9**). While there are potential trends in this dataset which suggest possible associations of exposures/phenotypes on mutation outcomes, the lack of significant statistical association and lack of concordance between different sperm sequencing panels makes it unclear if they are real. While we agree that the association of these exposures on mutations in sperm should continue to be investigated, we believe that it will require investigations in other (preferably larger) cohorts to confirm or disprove the lack of strong effects we see in this cohort. We have amended the text interpreting our findings to better reflect this (new text in italics):

Manuscript lines 360-363: “These results suggest that, unlike many somatic tissues, the male germline mutation landscape may be largely protected from these risk factors, *although further investigation from larger and more diverse cohorts will be needed to confirm this.*”

Rebuttal Fig. 9 | Phenotype effect sizes Effect size of cohort phenotypes to mutation outcome correlations, with different sequencing datasets indicated by colour. Joint predictor glm models used the gaussian family with FDR corrected P values. Asterisks indicate significance level of corrected P value: (*P value >0.01 to <0.05, **P value >0.001 to <0.01, ***P value <0.001). Error bars indicate 95% CIs of effect sizes.

R3.7 The manuscript does not discuss whether structural variants or copy-neutral loss-of-heterozygosity (LOH) events were observed in the dataset. This could be an interesting avenue to explore, given the implications for germline mutations.

We agree that it would be very interesting to explore this in sperm. Unfortunately there are technical limitations related to NanoSeq and sperm polyclonality which preclude this:

Structural variants: In short, the reason structural variants cannot be detected with NanoSeq is that during library preparation there is a low but significant rate of ligation between independent DNA fragments. Read pairs from these chimeric molecules map to different genomic regions and they often resemble a structural variant. Here, having duplex information (coverage of both strands) does not offer an error correction advantage. To call a structural variant with standard sequencing methods one typically relies on clonality, which translates in multiple independent reads supporting the same rearrangement. In sperm and other polyclonal samples sequenced with NanoSeq, clonal structural variants are not expected. Without such clonality, NanoSeq is unable to distinguish between true structural variants and the background rate of ligation between independent DNA fragments. Alternative approaches such as those that use long-read sequencing seem more promising for structural variants in sperm and indeed the paper (Schweiger et al. 2024), which we now point to in our discussion is able to detect these.

LOH events: Independent LOH events in highly polyclonal samples are not expected to result in biased biallelic frequencies and hence are again not detectable by NanoSeq. The only

exception we've considered where we believe that we may be able to detect this in the future is sequencing tissues of individuals who have a germline predisposition variant (e.g. a *BRCA1* allele mutated) where there is expected to be systematic LOH of the wild-type allele across many cells, resulting in a detectable level of biased biallelic frequencies of the same allele.

We hope that the paragraph added to our discussion mentioned in response to the reviewer's first comment related to STRs and segmental duplications conveys the potential of investigating these variant classes in the future.

R3.8 The findings should be explicitly linked to the 'selfish spermatogonial selection' hypothesis in the Discussion. How do these results refine or expand our understanding of the theory?

Thank you for this suggestion. Our findings strongly support the concept of positive selection in spermatogonia, and specifically many of the findings described in the 'selfish spermatogonial selection' papers of Goriely, Wilkie and colleagues. While we prefer the terminology of "positive selection" rather than "selfish selection" to emphasise its parallels with selection in other proliferative tissues, we have now added an explicit comparison with these previous findings to our discussion to better contextualise our work with the studies related to this theory.

Manuscript lines 423-436: "By analysing over 35,000 coding mutations from sperm exome-wide, we corroborate key findings from previous studies of spermatogonial selection while significantly advancing our understanding of its scope and impact. Our results replicate 9 of the 13 previously identified germline selection genes and align with findings that the impact of selection accumulates with age, leading to an increased fraction of sperm carrying pathogenic variants. Simultaneously, our work provides novel insights, identifying 31 additional genes under positive selection and broadening the range of pathways and mutational mechanisms implicated in this process. The discovery of diverse genes and the identification of loss-of-function as a common selection mechanism highlight the power of NanoSeq to enable exome-wide searches and to detect mutations in single cells, overcoming the limitations of previous methods reliant on high-frequency gain-of-function missense mutations. These findings also suggest that germline selection operates within the broader framework of cellular selection, driven by many of the same genes and mechanisms that shape clonal dynamics in somatic tissues. However, unlike somatic selection, which remains confined to the individual, germline-selected mutations can be inherited, impacting offspring phenotypes and influencing evolutionary trajectories."

R3.9 I would welcome a more extensive discussion of the bias against tumor suppressor mutations, as opposed to activating missense mutations, amongst positively selected genes. I could imagine that inactivated tumor suppressors may have a stronger effect across distinct cell types and tissues and thus more often result in embryonic viability, but suppose there may be other explanations too

Thanks for raising this point. We agree that there is a striking difference in the frequency of activating missense mutation genes vs. tumour suppressor mutation genes in previous works (13 vs. 0) vs this work (10 vs. 30). Several possible expansions come to mind to explain this observation:

1- technical limitations: previous studies relied on targeted approaches which mostly cover highly recurrent mutations, particularly activation missense hotspots. Hence, given the technical challenges and detection limits with these approaches, these studies were mostly focused on sites that could be identified at very high frequencies in individuals with disease. With the NanoSeq method, we are now able to detect mutations at single-molecule accuracy and survey across a much larger breadth of the genome, which enabled us to identify LOFs that may be less recurrent but still confer selective advantage in spermatogonia stem cells. We have added a sentence to our discussion addressing this point (also shown in context in the paragraph above):

Manuscript lines 429-432: “The discovery of diverse genes and the identification of loss-of-function as a common selection mechanism highlight the power of NanoSeq to enable exome-wide searches and to detect mutations in single cells, overcoming the limitations of previous methods reliant on high-frequency gain-of-function missense mutations.”

There are also other possible explanations which we may contribute that we speculate on here and would be interesting avenues for future explorations:

2- Activating missense mutations often produce highly specific gain of function phenotypes that directly promote clonal expansions. While, inactivated tumour suppressors may have broader effects that influence cell division, survival, or differentiation pathways across multiple tissues/cell types. This broader impact could in principle increase the likelihood of embryonic lethality when these mutations occur at higher frequencies, which may explain their relative rarity in early experimental datasets that focused on high recurrent mutations.

3- Another possible explanation is, LOFs may operate through distinct mechanisms compared to GOFs. For example, tumour suppressor mutations may provide a subtler selective advantage, particularly in the context of aging, causing slower clonal growth and expansion without triggering strong negative selection. This may also explain why early studies may not have sufficient power to identify small to modest size clonal expansions, and now we can uncover them with a more sensitive sequencing approach.

R3.10 A more detailed discussion of potential clinical implications of the findings, particularly for counseling older prospective fathers, would enhance the manuscript's impact.

Thanks for this suggestion, we agree that potential clinical implications for our findings would be of great interest. However, although we have quantified the increased risk of pathogenic *de novo* mutations in sperm with age, translating this information into potential actionable clinical recommendations presents challenges. While the risk for older fathers is undoubtedly higher than that of younger fathers, it is important to note that the absolute risk of serious outcomes for offspring remains relatively low. Additionally, the mutations we observe are diverse and consistently at extremely low allele frequencies, making any potential screening applications difficult to currently conceive of. Hence, our interpretation is that there are currently limited practical steps that older prospective fathers can take based on our current results beyond being informed of the elevated risks. We have now included text on this topic in the discussion:

Manuscript lines 483-499:

“Nevertheless, growing awareness of these risks is likely to increase interest in how this knowledge could inform reproductive decisions, genetic counselling, or clinical interventions. While it's clear that disease risks for children increase with paternal age, translating these findings into specific clinical recommendations presents challenges. Unlike screening for familial transmitted mutations or aneuploidies, common events already integrated into reproductive medicine, screening for pathogenic variants from sperm is more difficult. These mutations occur at extremely low individual allele frequencies, yet collectively pose a significant disease risk. As a result, an effective screening assay would need to test a vast number of mutations with exceptionally high accuracy to maintain an acceptable false-positive rate.

Additionally, the relationship between sperm mutations and birth prevalence remains uncertain. Many pathogenic variants in sperm may not result in live births due to impaired fertilisation, embryonic lethality, or increased pregnancy loss (Supplementary Note 4). Nonetheless, targeted risk assessment may be particularly valuable in certain scenarios. For instance, individuals with an elevated risk of sperm hypermutation, either due to intrinsic factors such as impaired DNA replication or repair, or extrinsic exposures like chemotherapy, may benefit from pre-screening tests or proactive reproductive planning (Kaplanis et al 2020, Santarsieri et al. 2025).”

Issues of minor concern:

R3.11 The sentence in the abstract: “Most positively selected genes are associated with developmental or cancer predisposition disorders in children, while four genes that exhibit elevated frequencies of protein-truncating variants in healthy populations. ”is awkwardly phrased. Please improve clarity.

Removed “that” (Manuscript Line 24).

R3.12 The statement that “spermatogonial stem cells are the only replicating cells with the potential to transmit mutations to offspring” seems incorrect to me. Recent work (e.g., Porubsky et al., 2024; <https://www.biorxiv.org/content/10.1101/2024.08.05.606142v1> [biorxiv.org]), highlights that transmitted mutations may also arise post-zygotically in replicating cells.

We see where the reviewer is coming from as DNMs are generally defined as transmitted mutations and a portion of DNMs are postzygotic mutations. However we would argue that if they are postzygotic then by definition they are after the zygote and not yet transmitted from one generation to its offspring. In the attached manuscript this is described directly “*We classify 17.1% (129/755) of de novo SNVs as PZMs, defined here as somatic mutations occurring very early in development.*” In this sense it is debatable whether PZMs should really be called DNMs for that generation and should instead only be called a DNM if they make it to the “3rd” generation - but this is a bit of a digression. The intended message we are conveying is that of all the regularly replicating cell types in the parental body - only mutations in spermatogonia show up in a zygote.

R3.13 Fig. 2abc could benefit from clearer labeling, and Fig. 3c should indicate whether observed age trends are statistically significant.

We have added a second gene label along with Fig 2c (please see below) which we hope clarifies the labelling.

For Fig. 3c the expected line was created with age as a linear input variable so would not be valid to test its significance, but the observed relationship is indeed significant as added to the text:

Manuscript lines 296-297: "... with a significant relationship between the observed fraction and age ($P = 1.75e-05$; Fig. 3c)."

Modified manuscript main Figure 2, added a second gene label to panel c

R3.14 Are sperm counts particularly linked with the number of mutations acquired in sperm?

This answer is partially addressed at the end of **Supplementary Note 1** where we show that excluding oligospermic samples doesn't have an impact on our mutation burden estimates:

"The mutation burden estimate in sperm presented in **Figure 1** of the main text was 1.67 haploid single base substitution (SBS) per year (95% confidence interval (CI) = 1.41-1.92, linear mixed-effect regression) with an intercept of 3.0 haploid SBS (95% CI = -10.3 to 16.4). This estimate is not substantially impacted by excluding samples with oligozoospermia: 1.68 haploid SBS per year (95% CI = 1.38-1.98, linear mixed-effect regression) and intercept of 3.1 haploid SBS (95% CI = -12.2 to 18.5), excluding those samples with a ratio of sperm to non-sperm cells below ten: 1.68 haploid SBS per year (95% CI = 1.41-1.94, poisson regression) and intercept of 3.4 haploid SBS (95% CI = -10.2 to 17.1) ..."

A more direct test of the relationship between these variables finds a similar answer, where we find no significant relationships between sperm count and any mutation burden variables in any of our sperm sequencing sets (**Rebuttal Fig. 10**).

Rebuttal Fig. 10 | Sperm count vs burdens Effect size of sperm count and age to mutation outcome correlations, with different sequencing datasets indicated by colour. Joint predictor glm models used the gaussian family with FDR corrected P values. Asterisks indicate significance level of corrected P value: (*P value >0.01 to <0.05, **P value >0.001 to <0.01, ***P value <0.001). Error bars indicate 95% CIs of effect sizes.

References

- Abascal F, Harvey LMR, Mitchell E, Lawson ARJ, Lensing SV, Ellis P, Russell AJC, Alcantara RE, Baez-Ortega A, Wang Y, et al. 2021. Somatic mutation landscapes at single-molecule resolution. *Nature* **593**: 405–410.
- Aggarwala V, Voight BF. 2016. An expanded sequence context model broadly explains variability in polymorphism levels across the human genome. *Nat Genet* **48**: 349–355.
- Axelsson J, LeBlanc D, Shojaeisaadi H, Meier MJ, Fitzgerald DM, Nachmanson D, Carlson J, Golubeva A, Higgins J, Smith T, et al. 2024. Frequency and spectrum of mutations in human sperm measured using duplex sequencing correlate with trio-based de novo mutation analyses. *Sci Rep* **14**: 23134.
- Bernstein N, Spencer Chapman M, Nyamondo K, Chen Z, Williams N, Mitchell E, Campbell PJ, Cohen RL, Nangalia J. 2024. Analysis of somatic mutations in whole blood from 200,618 individuals identifies pervasive positive selection and novel drivers of clonal hematopoiesis. *Nat Genet* 1–9.
- Brunner SF, Roberts ND, Wylie LA, Moore L, Aitken SJ, Davies SE, Sanders MA, Ellis P, Alder C, Hooks Y, et al. 2019. Somatic mutations and clonal dynamics in healthy and cirrhotic human liver. *Nature* **574**: 538–542.
- Carlson J, Locke AE, Flickinger M, Zawistowski M, Levy S, Myers RM, Boehnke M, Kang HM, Scott LJ, Li JZ, et al. 2018. Extremely rare variants reveal patterns of germline mutation rate heterogeneity in humans. *Nat Commun* **9**: 3753.

- Choi S-K, Yoon S-R, Calabrese P, Arnheim N. 2012. Positive Selection for New Disease Mutations in the Human Germline: Evidence from the Heritable Cancer Syndrome Multiple Endocrine Neoplasia Type 2B. *PLOS Genet* **8**: e1002420.
- Eboreime J, Choi S, Yoon S, Sadybekov A, Katritch V, Calabrese P, Arnheim N. 2022. Germline selection of PTPN11 (HGNC:9644) variants make a major contribution to both Noonan syndrome's high birth rate and the transmission of sporadic cancer variants resulting in fetal abnormality. *Hum Mutat* **43**: 2205–2221.
- Goriely A, McVean GAT, Røjmyr M, Ingemarsson B, Wilkie AOM. 2003. Evidence for Selective Advantage of Pathogenic FGFR2 Mutations in the Male Germ Line. *Science* **301**: 643–646.
- Jaiswal S, Fontanillas P, Flannick J, Manning A, Grauman PV, Mar BG, Lindsley RC, Mermel CH, Burt N, Chavez A, et al. 2014. Age-related clonal hematopoiesis associated with adverse outcomes. *N Engl J Med* **371**: 2488–2498.
- Karczewski KJ, Francioli LC, Tiao G, Cummings BB, Alföldi J, Wang Q, Collins RL, Laricchia KM, Ganna A, Birnbaum DP, et al. 2020. The mutational constraint spectrum quantified from variation in 141,456 humans. *Nature* **581**: 434–443.
- Kunisaki J, Goldberg ME, Lulla S, Sasani T, Hiatt L, Nicholas TJ, Liu L, Torres-Arce E, Guo Y, James E, et al. 2024. Sperm from infertile, oligozoospermic men have elevated mutation rates. <http://medrxiv.org/lookup/doi/10.1101/2024.08.22.24312232> (Accessed September 2, 2024).
- Lawson ARJ, Abascal F, Coorens THH, Hooks Y, O'Neill L, Latimer C, Raine K, Sanders MA, Warren AY, Mahbubani KTA, et al. 2020. Extensive heterogeneity in somatic mutation and selection in the human bladder. *Science* **370**: 75–82.
- Lawson ARJ, Abascal F, Nicola PA, Lensing SV, Roberts AL, Kalantzis G, Baez-Ortega A, Brzozowska N, Moustafa JSE-S, Vaitkute D, et al. 2024. Somatic mutation and selection at epidemiological scale. 2024.10.30.24316422. <https://www.medrxiv.org/content/10.1101/2024.10.30.24316422v2> (Accessed January 29, 2025).
- Lee-Six H, Olafsson S, Ellis P, Osborne RJ, Sanders MA, Moore L, Georgakopoulos N, Torrente F, Noorani A, Goddard M, et al. 2019. The landscape of somatic mutation in normal colorectal epithelial cells. *Nature* **574**: 532–537.
- Liu MH, Costa BM, Bianchini EC, Choi U, Bandler RC, Lassen E, Grońska-Pęski M, Schwing A, Murphy ZR, Rosenkjær D, et al. 2024. DNA mismatch and damage patterns revealed by single-molecule sequencing. *Nature* 1–10.
- Lundström O (Sachenkova), Adriaan Verbiest M, Xia F, Jam HZ, Zlobec I, Anisimova M, Gymrek M. 2023. WebSTR: A Population-wide Database of Short Tandem Repeat Variation in Humans. *J Mol Biol* **435**: 168260.
- Maher GJ, Goriely A, Wilkie AOM. 2014. Cellular evidence for selfish spermatogonial selection in aged human testes. *Andrology* **2**: 304–314.
- Maher GJ, McGowan SJ, Giannoulatou E, Verrill C, Goriely A, Wilkie AOM. 2016. Visualizing the origins of selfish de novo mutations in individual seminiferous tubules of human testes. *Proc Natl Acad Sci* **113**: 2454–2459.

- Martincorena I, Fowler JC, Wabik A, Lawson ARJ, Abascal F, Hall MWJ, Cagan A, Murai K, Mahbubani K, Stratton MR, et al. 2018. Somatic mutant clones colonize the human esophagus with age. *Science* **362**: 911–917.
- Martincorena I, Roshan A, Gerstung M, Ellis P, Van Loo P, McLaren S, Wedge DC, Fullam A, Alexandrov LB, Tubio JM, et al. 2015. High burden and pervasive positive selection of somatic mutations in normal human skin. *Science* **348**: 880–886.
- Mitchell E, Spencer Chapman M, Williams N, Dawson KJ, Mende N, Calderbank EF, Jung H, Mitchell T, Coorens THH, Spencer DH, et al. 2022. Clonal dynamics of haematopoiesis across the human lifespan. *Nature* **606**: 343–350.
- Moore L, Leongamornlert D, Coorens THH, Sanders MA, Ellis P, Dentre SC, Dawson KJ, Butler T, Rahbari R, Mitchell TJ, et al. 2020. The mutational landscape of normal human endometrial epithelium. *Nature* **580**: 640–646.
- Sasani TA, Pedersen BS, Gao Z, Baird L, Przeworski M, Jorde LB, Quinlan AR. 2019. Large, three-generation human families reveal post-zygotic mosaicism and variability in germline mutation accumulation eds. A.L. Williams, M.I. McCarthy, and A.L. Williams. *eLife* **8**: e46922.
- Schweiger R, Lee S, Zhou C, Yang T-P, Smith K, Li S, Sanghvi R, Neville M, Mitchell E, Nessa A, et al. 2024. Insights into non-crossover recombination from long-read sperm sequencing. 2024.07.05.602249.
<https://www.biorxiv.org/content/10.1101/2024.07.05.602249v1> (Accessed January 29, 2025).
- Seplyarskiy V, Moldovan MA, Koch E, Kar P, Neville MD, Rahbari R, Sunyaev S. 2025. Cohort-level analysis of human de novo mutations points to drivers of clonal expansion in spermatogonia. 2025.01.03.25319979.
<https://www.medrxiv.org/content/10.1101/2025.01.03.25319979v1> (Accessed January 29, 2025).
- Shojaeisaadi H, Schoenrock A, Meier MJ, Williams A, Norris JM, Palmer ND, Yauk CL, Marchetti F. 2024. Mutational signature analyses in multi-child families reveal sources of age-related increases in human germline mutations. *Commun Biol* **7**: 1–12.
- Sondka Z, Bamford S, Cole CG, Ward SA, Dunham I, Forbes SA. 2018. The COSMIC Cancer Gene Census: describing genetic dysfunction across all human cancers. *Nat Rev Cancer* **18**: 696–705.
- Tan X, Zheng C, Zhuang Y, Jin P, Wang F. 2023. The m6A reader PRRC2A is essential for meiosis I completion during spermatogenesis. *Nat Commun* **14**: 1636.
- Wood KA, Goriely A. 2022. The impact of paternal age on new mutations and disease in the next generation. *Fertil Steril*.
<https://www.sciencedirect.com/science/article/pii/S0015028222019677> (Accessed November 24, 2022).
- Yoshida S. 2020. Mouse Spermatogenesis Reflects the Unity and Diversity of Tissue Stem Cell Niche Systems. *Cold Spring Harb Perspect Biol* **12**: a036186.

Response to Referees Letter

The original reviewers' comments and questions are in blue, while our responses are in black.

Referee #2 (Remarks to the Author):

The authors did an excellent job putting together the response to reviewers, which adequately addressed all my major questions. Below are some minor points that I hope they could consider incorporating in the manuscript:

1. Mutation calling accuracy (previously raised by reviewer1)

Although Fig 1 and Rebuttal Fig 1 clearly show that the mutation burden, as a function of age/sample, are highly concordant across cohorts/methods, these results do not directly speak to the accuracy of individual mutation calls, as it is possible that false positives are roughly balanced by a similar number of false negatives, yet the rates of both are substantial. Since detection of positive selection is reliant on the exact mutation calls rather than the mutation burden per sample, it is important to demonstrate that the individual mutation calls are highly reliable. It will be helpful to include an estimate of the false discovery rate of NanoSeq (I understand FNR is more challenging to estimate).

Added "...with an error rate $< 5 \times 10^{-9}$ per bp" to the introduction to specify the proven FDR of NanoSeq as suggested

2. Sensitivity to mutation model

I understand and agree with the authors' argument that training mutation model with longer sequence context and finer methylation bins are infeasible given current data and the resulting changes in the results may be marginal. However, a general point I wanted to make is that the positive selection test relies on the null mutation model, and inference of the mutation model always makes certain assumptions and is still rapidly improving. Therefore, the specific dN/dS values (exome-wide or for sets of genes) may be subject to revision with further improved mutation models that become available in the future (although I have little doubt that most of the candidate genes are robust to the baseline mutation model). I hope the authors can add a brief comment on this in the description of results, or in the Discussion.

We have added the suggested text to the supplement: "It is reasonable to assume that as mutation models continue to improve and training data sources grow, the dN/dS estimates presented here may be subject to revision in the future."

3. In lines 40-41, "replicating cells" can perhaps be better phrased as "proliferating cells" or "dividing cells". It will also be helpful to add a qualifier "in the adult body" to these sentences, which will address the comment of reviewer 3 on postzygotic mutations (which I believe reviewer 3 meant for early embryonic mutations in the parent; these mutations technically arise in dividing cells and can be indeed transmitted to offspring).

Changed wording of these main text lines to "adult proliferating" to satisfy both suggestions.